# Type III CRISPR-Cas systems can provide redundancy to counteract viral escape from type I systems

Sukrit Silas[1,2], Patricia Lucas-Elio[3], Simon A Jackson[4], Alejandra Aroca-Crevillén[3], Loren L Hansen[1], Peter C Fineran[4,5], Andrew Z Fire[1]*, Antonio Sánchez-Amat[3]*

[1]Department of Pathology, Stanford University, Stanford, United States; [2]Department of Chemical and Systems Biology, Stanford University, Stanford, United States; [3]Department of Genetics and Microbiology, Universidad de Murcia, Murcia, Spain; [4]Department of Microbiology and Immunology, University of Otago, Dunedin, New Zealand; [5]Bio-Protection Research Centre, University of Otago, Dunedin, New Zealand

**Abstract** CRISPR-Cas-mediated defense utilizes information stored as spacers in CRISPR arrays to defend against genetic invaders. We define the mode of target interference and role in antiviral defense for two CRISPR-Cas systems in *Marinomonas mediterranea*. One system (type I-F) targets DNA. A second system (type III-B) is broadly capable of acquiring spacers in either orientation from RNA and DNA, and exhibits transcription-dependent DNA interference. Examining resistance to phages isolated from Mediterranean seagrass meadows, we found that the type III-B machinery co-opts type I-F CRISPR-RNAs. Sequencing and infectivity assessments of related bacterial and phage strains suggests an 'arms race' in which phage escape from the type I-F system can be overcome through use of type I-F spacers by a horizontally-acquired type III-B system. We propose that the phage-host arms race can drive selection for horizontal uptake and maintenance of promiscuous type III interference modules that supplement existing host type I CRISPR-Cas systems.
DOI: https://doi.org/10.7554/eLife.27601.001

*For correspondence: afire@stanford.edu (AZF); antonio@um.es (ASá-A)

Competing interests: The authors declare that no competing interests exist.

## Introduction

CRISPR-Cas systems enable adaptive immunity in prokaryotes in three stages (*Barrangou et al., 2007*). First, molecular memories of infection are formed by the acquisition of short segments of foreign nucleic acids, which are stored as 'spacers' in CRISPR arrays (*adaptation*) (*Deveau et al., 2008*; *Jackson et al., 2017*). Second, these CRISPR arrays are transcribed into precursor transcripts, then processed into CRISPR RNAs (crRNA *processing*) (*Brouns et al., 2008*; *Hochstrasser and Doudna, 2015*). Finally, the crRNAs form complexes with CRISPR-associated (Cas) proteins that execute a variety of protective responses against matching nucleic acid targets (*interference*) (*Plagens et al., 2015*). Thus, CRISPR-Cas systems function as sequence-specific nucleases that degrade invasive genetic elements.

CRISPR-Cas loci have been organized into six phylogenetic types (*Makarova et al., 2015*; *Shmakov et al., 2015*). All characterized CRISPR-Cas systems utilize crRNAs for sequence-specific identification of targets. However, different types exhibit varying modes of target interference. Type I, type II, and type V systems use crRNAs to guide the recognition and destruction of DNA targets (*Barrangou et al., 2007*; *Brouns et al., 2008*; *Zetsche et al., 2015*), whereas crRNA-guided recognition of RNA targets occurs in type VI systems (*Abudayyeh et al., 2016*). In type III systems, the interference complex recognizes nascent RNA transcripts containing the reverse complement of the crRNA sequence, and degrades both the transcript and its template DNA in a process termed

transcription-dependent DNA interference (*Marraffini and Sontheimer, 2008*; *Hale et al., 2009*; *Hale et al., 2012*; *Deng et al., 2013*; *Goldberg et al., 2014*; *Tamulaitis et al., 2014*; *Peng et al., 2015*; *Samai et al., 2015*). Thus, interference can only occur if type III crRNAs, and hence spacers, are antisense to their cognate RNA targets. CRISPR transcription is unidirectional; as such, new type III spacers must be inserted into CRISPR arrays in a specific orientation to produce crRNAs that are functional in interference.

Other CRISPR-Cas types also require integration of new spacers in one specific orientation. Type I, II, and V CRISPR-Cas systems rely on the presence of a protospacer adjacent motif (PAM) to distinguish foreign DNA targets from the host-copy of the spacer, thereby preventing lethal self-targeting of the host CRISPR loci (*Mojica et al., 2009*; *Stern et al., 2010*; *Westra et al., 2013*). A consequence of PAM-based target authentication is that spacers must be oriented such that the PAM-proximal ends of crRNAs are positioned correctly with respect to the PAM-sensing domain of the interference complex. During CRISPR adaptation in these systems, a PAM-sensing domain in the Cas1-Cas2 spacer acquisition complex ensures spacers are integrated in the correct orientation (*Jackson et al., 2017*; *Shipman et al., 2016*; *Wang et al., 2015*; *Nuñez et al., 2015*). In contrast, type III CRISPR-Cas systems typically do not rely on PAMs to prevent self-targeting, although a highly degenerate RNA sequence constraint (rPAM) has been reported in *P. furiosus* (*Elmore et al., 2016*). Instead, base-pairing between the CRISPR repeat-derived 5' crRNA handle and the protospacer flanking sequence has been proposed to suppress target recognition, thereby preventing self-targeting in the context of the host CRISPR array (*Marraffini and Sontheimer, 2010*). Whether, and how, spacers are oriented in type III CRISPR arrays remains unclear.

Type III CRISPR-Cas systems comprise a large and diverse group, divided into 4 subtypes (III-A, B, C, and D). Type III-A systems usually possess spacer acquisition and pre-crRNA processing factors (Cas1-Cas2 and Cas6 homologues, respectively), whereas many type III-B, C, and D systems lack these components (*Makarova et al., 2015*). Such systems commonly occur in genomes containing type I CRISPR-Cas loci (*Makarova et al., 2011*), and in many cases share 'communal' CRISPR arrays utilizing the same repeat sequences as the co-habiting type I systems. In support of this, co-immuno-precipitation experiments have confirmed the association of 'shared' crRNAs with type III interference complexes (*Staals et al., 2013*; *Staals et al., 2014*; *Elmore et al., 2015*; *Majumdar et al., 2015*). These associations can also lead to transcription-dependent interference against protospacer-containing plasmids in *Sulfolobus islandicus* REY15A (*Deng et al., 2013*) and *Pyrococcus furiosus* (*Elmore et al., 2016*). There are no type III-specific CRISPR spacer arrays in the *S. islandicus* and *P. furiosus* genomes; such type III interference modules have thus been proposed to stably co-reside with type I CRISPR loci, and to depend on type I factors for spacer acquisition and crRNA maturation (*Makarova et al., 2015*). While the frequent co-habitation of type III with type I systems could provide benefits to the host (*Deng et al., 2013*; *Elmore et al., 2016*), the mechanistic underpinning of this evolutionary association is unknown. Moreover, the biological and ecological role(s) of interactions between evolutionarily disparate CRISPR-Cas systems remain unexplored.

Recently, we showed that the *Marinomonas mediterranea* MMB-1 type III-B system uses a reverse transcriptase (RT)-Cas1 fusion protein to acquire spacers directly from RNA molecules (*Silas et al., 2016*). The *M. mediterranea* genome contains a type I-F and a type III-B system, each with a full complement of spacer acquisition, processing, and interference components. Furthermore, each system possesses its own CRISPR loci, with distinct repeat sequences. Here, we show that the *M. mediterranea* type III-B effector complex naturally utilizes crRNAs from both type III-B and type I-F CRISPR loci for interference, despite their divergent CRISPR repeat sequences. We demonstrate that this surprising plasticity in crRNA selection allows the type III-B interference machinery to use type I-F spacers to counter infection from natural *M. mediterranea* phages abundant in the native host ecosystem, which have escaped the type I-F defenses through genetic (PAM) mutations. Based on the analysis of the sequences of highly related bacterial strains, we propose that some type III-B systems exhibiting plasticity in their crRNA utilization criteria behave as promiscuous modules. This flexibility in crRNA usage might serve as an important trait for their selection following horizontal transfer into bacterial hosts that are challenged by phages that had escaped their type I CRISPR-Cas defenses.

# Results

## Transcription-dependent degradation of DNA by the *M. mediterranea* type III-B CRISPR-Cas system

The *M. mediterranea* MMB-1 genome contains type III-B and type I-F CRISPR-Cas systems (*Figure 1A*). We examined the interference mechanism for the type III-B system using a conjugation-based plasmid interference assay. Each of the plasmids tested contained a target sequence matching either the first or second native spacer from the type III-B CRISPR array (CRISPR03) inserted in either the sense or antisense orientation relative to a constitutive promoter ($P_{amp}$). We flanked the target sequences with either the native CRISPR repeats or randomized sequences with identical base composition (*Figure 1B*). Consistent with transcription-dependent DNA interference, the conjugation efficiency was reduced only when the reverse complements of the spacer sequences were transcribed from the plasmid (*Figure 1C*; also see *Figure 1—source data 1*). As expected for type III self vs non-self discrimination, CRISPR repeat sequences flanking the target protected against interference (*Marraffini and Sontheimer, 2010*).

To test the hypothesis that the observed interference was due to type III-B activity on DNA, we repeated the assays in several *M. mediterranea* mutants. Interference was not detected when genes encoding the type III-B effector complex were removed (Δ*cmr*) or when the entire type III-B operon was deleted (ΔIII-B) (*Figure 1C*). Additional mutational analysis indicated roles for both *cmr2* and *cmr4* in interference. Similarly, mutation of the conserved GGDD motif (Cmr2:GGAA) within the PALM domain in the Cas10-family member Cmr2 abolished interference (*Figure 1C*), as observed in other systems (*Hatoum-Aslan et al., 2014*). We also tested the requirement for a conserved Asp residue in Cmr4 that has been reported to be necessary for target RNA degradation, but not for DNA interference (*Samai et al., 2015*). In agreement, the *M. mediterranea* Cmr4 D26A mutant was capable of transcription-dependent DNA interference (*Figure 1C*; also see *Figure 1—figure supplement 1*).

To examine whether the type III-B system was dependent on the type I-F locus, we tested type I-F deletion mutants. Removal of the type I-F operon (ΔI-F) had no effect on interference of the plasmids targeted by the type III-B system. Likewise, deletion of the type I-F CRISPR array (ΔI-F CRISPR04), or deletion of the spacer acquisition and effector nuclease genes *cas1* and *cas2-3* (ΔI-F Cas123) had no effect. Simultaneous deletion of both type I-F and III-B operons abrogated plasmid interference (*Figure 1C*). Our results demonstrate that the *M. mediterranea* type III-B system facilitates transcription-dependent target DNA interference with no requirement for the type I-F locus.

## Promiscuous capacity for spacer acquisition: absence of a concerted mechanism to orient new type III spacers

The transcription-dependent DNA interference we observed for the type III-B system implied that, to be functional, spacers would need to be inserted into the CRISPR array in a specific direction, such that crRNAs are complementary to their RNA targets. We previously showed that overexpression of the type III-B CRISPR adaptation components (*Figure 2A*) in *M. mediterranea* can facilitate spacer acquisition from cellular RNAs and DNAs (*Silas et al., 2016*). In wild type, the new spacers appear to be predominantly acquired from RNA, as evident from the transcriptional bias observed with over-expression of RT-Cas1 but not RTΔ RT-Cas1 (*Figure 2—figure supplement 1*) – the role of RNA as the source of spacers, rather than transcribed DNA, was deduced previously (*Silas et al., 2016*). However, the spacers acquired with WT RT-Cas1 showed a bias toward the orientation that would result in the generation of non-functional crRNAs that were not complementary to source mRNAs (*Figure 2B*). We reasoned that the observed bias could be a consequence of autoimmunity, as spacers yielding crRNAs complementary to cellular transcripts would be lethal (*Stern et al., 2010*; *Vercoe et al., 2013*). Thus, any mechanism for the acquisition of functional spacers (that are antisense to host mRNAs) would likely be masked by the presence of the native type III-B interference module.

Consistent with this, the ΔIII-B and Δ*cmr* mutants displayed RNA-derived spacer acquisition with a moderate bias toward spacers targeting the antisense strand of target genes (*Figure 2B*; *Figure 2—figure supplement 1*), that is, the presence of the type III-B targeting module did indeed skew detection of the acquisition of antisense spacers in the assays with wildtype (WT). Despite the clear

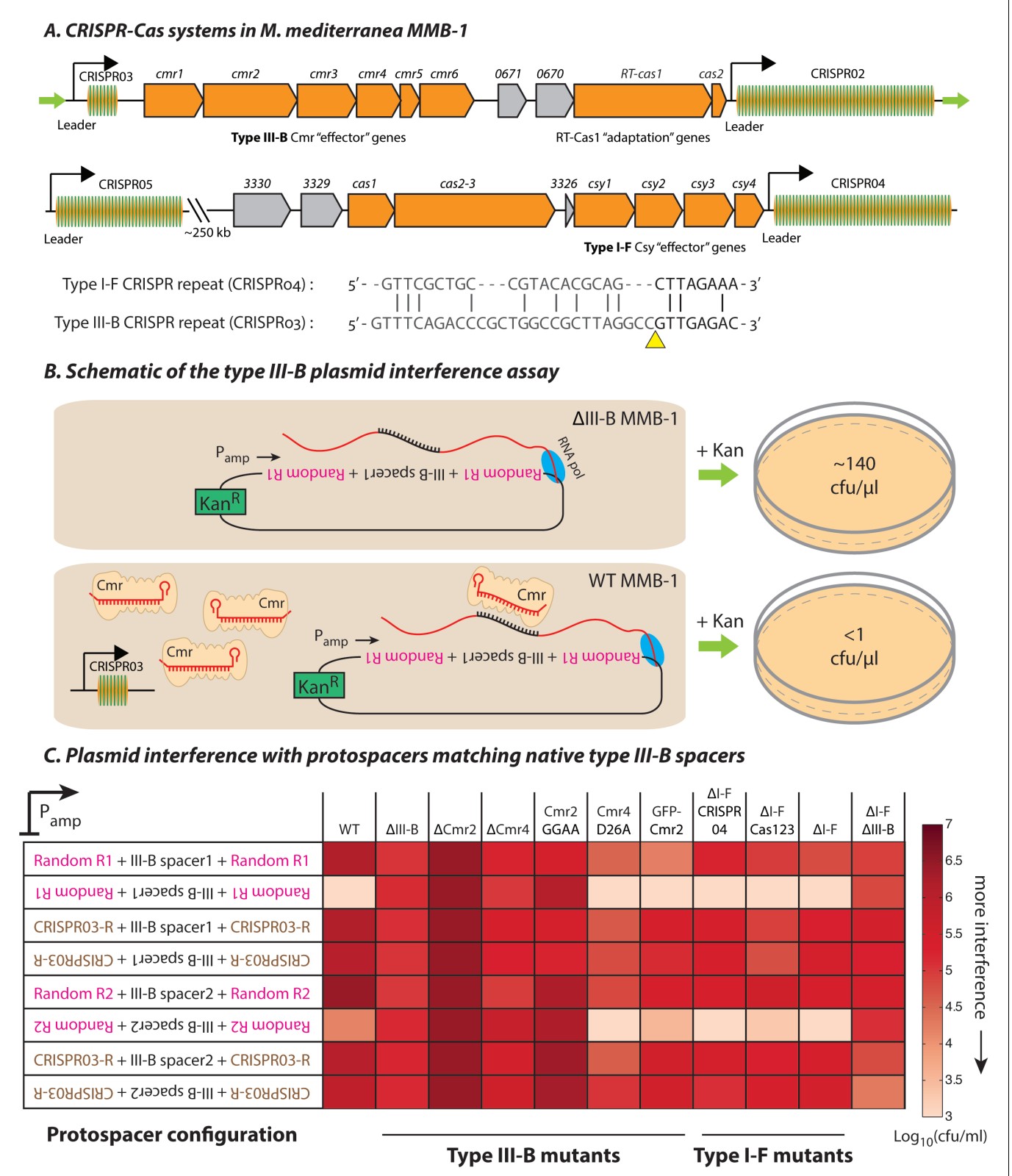

**Figure 1.** Transcription-dependent plasmid elimination by the *M. mediterranea* type III-B CRISPR-Cas system. (**A**) Schematics of the type III-B (above; adapted from [*Silas et al., 2016*]) and type I-F (below) CRISPR-Cas systems in *M. mediterranea* MMB-1. The type III-B operon contains genes encoding the adaptation proteins RT-Cas1 and Cas2, a 6-gene *cmr* cassette encoding the type III-B effector complex, the 58-spacer CRISPR02 array, the 8-spacer CRISPR03 array, and two genes of unknown function (Marme_0670, 0671). The Cas10 HD nuclease domain, required for DNA interference in some type

*Figure 1 continued on next page*

*Figure 1 continued*

III-B systems (**Elmore et al., 2016**), is not evident in the *M. mediterranea* Cas10 (Cmr2). The entire operon is flanked by two ~200 bp direct repeats (green arrows). The type I-F system contains *cas1, cas2-3,* four *csy* genes and a 52 spacer CRISPR04 array. Marme_3330, 3329, and 3326 are ORFs of unknown function. The 37-spacer CRISPR05 array, located ~250 kb away from the type I-F system has a near-canonical type I-F repeat sequence but no neighboring *cas* genes. An alignment of predicted 5'-crRNA handles (last 8 nt) from type I-F and type III-B CRISPR repeats is shown, with the remaining repeat nucleotides aligned separately (grey) and the crRNA processing sites depicted by the yellow triangle. (**B**) Overview of the plasmid interference assay using native CRISPR03 spacers. Configurations that allow the protospacer RNA to be targeted by the corresponding endogenous crRNA-Cmr complex lead to plasmid elimination and cell death on selective medium (+Kan). Inactivation of the type III-B CRISPR-Cas system results in plasmid stability and a corresponding increase in transconjugants. Numbers shown correspond to colony forming units per microliter (cfu/µl) measurements from the conjugations depicted. (**C**) Transcription-dependent plasmid interference in various *M. mediterranea* mutants using native CRISPR03 spacers. Log-transformed cfu/ml measurements from conjugations of plasmids with different protospacer configurations (relative to P$_{amp}$). Random R1 and R2 (pink text) are two different randomized sequences with base composition identical to the type III-B CRISPR repeat (brown text; CRISPR03-R). Upside down text denotes the reverse-complement. The GFP-Cmr2 strain is a control for the process of mutant construction.

DOI: https://doi.org/10.7554/eLife.27601.002

The following source data and figure supplement are available for figure 1:

**Source data 1.** Colony forming units per mL (cfu/ml) counts obtained from conjugations for plasmid interference assays.
DOI: https://doi.org/10.7554/eLife.27601.004

**Figure supplement 1.** Effect of Cmr2 and Cmr4 active site mutations on protospacer RNA levels during putative type III-B RNA targeting.
DOI: https://doi.org/10.7554/eLife.27601.003

shift in bias in the absence of type III-B interference, antisense spacers did not constitute the overwhelming majority; arguing against a strong, concerted mechanism to orient incoming spacers into the CRISPR array. Moreover, the strand bias entirely disappeared when the RT-deleted version of RT-Cas1 (RTΔ RT-Cas1) was supplied (**Figure 2B**), thereby highlighting an intrinsic difference between spacer acquisition from RNA and DNA sources (RTΔ RT-Cas1 has previously been shown to acquire spacers only from DNA [**Silas et al., 2016**]). Finally, we tested whether the strand bias of newly acquired spacers in other CRISPR-Cas mutants was consistent with their interference capabilities. With over-expressed WT RT-Cas1, all strains supported spacer acquisition from host RNAs (**Figure 2—figure supplement 2**). As expected, the mutants capable of plasmid interference showed a bias for spacers matching the sense strand of source RNAs. Conversely, mutants incapable of interference showed a moderate bias for spacers matching the reverse complement of the source RNA (**Figure 2C**). In summary, the orientation of new type III-B spacers bears only a partial relationship to the strandedness of the source transcript. Additionally, host-derived spacers integrated in the antisense orientation are negatively selected in the presence of type III-B interference, due to transcription-dependent autoimmune self-targeting.

## *M. mediterranea* type I-F spacers provide protection against an abundant marine phage

To evaluate type III-B targeting in an ecological context, we isolated *M. mediterranea* phages from the original source of the MMB-1 strain in seawater from *Posidonia oceanica* seagrass meadows off the Mediterranean coast in south-eastern Spain. An enrichment culture from environmental samples from the coast of Cabo de Palos yielded plaques on lawns of the ΔIII-B *M. mediterranea* strain. The genomes of two phages, CPG1g and CPP1m (which produced plaques with different sizes) were sequenced, assembled, and annotated. The two ~44 kb genomes differ at only 7 sites, indicating that they are polymorphic variants of the same virus (also see Materials and methods). An additional 25 phage isolates were confirmed as likely variants of CPG1g using PCR probes designed against conserved regions in phage genes (data not shown). *Marinomonas* phage CPG1g has a 44,244 bp double stranded DNA genome. It contains 50 predicted open reading frames and one tRNA (Arg). Phylogenetic analyses based on the DNA polymerase protein sequence revealed that CPG1g belongs to the SP6 cluster within the T7 supercluster of the *Podoviridae* family and is most closely related to *Vibrio* phage Vc1 (**Li et al., 20152016**) and Ø318 (**Liu et al., 2014**) (**Figure 3—figure supplement 1**). The only *Marinomonas* phage described previously belongs to the *Siphoviridae* family (**Kang et al., 2012**). In agreement with a classification as a podovirus, electron microscopy revealed that CPG1g/CPP1m possess a very short tail (**Figure 3A**). Other *M. mediterranea* strains (IVIA-Po-186 (**Espinosa et al., 2010**) and CPR1 (this study)) are sensitive to CPG1g infection, whereas the

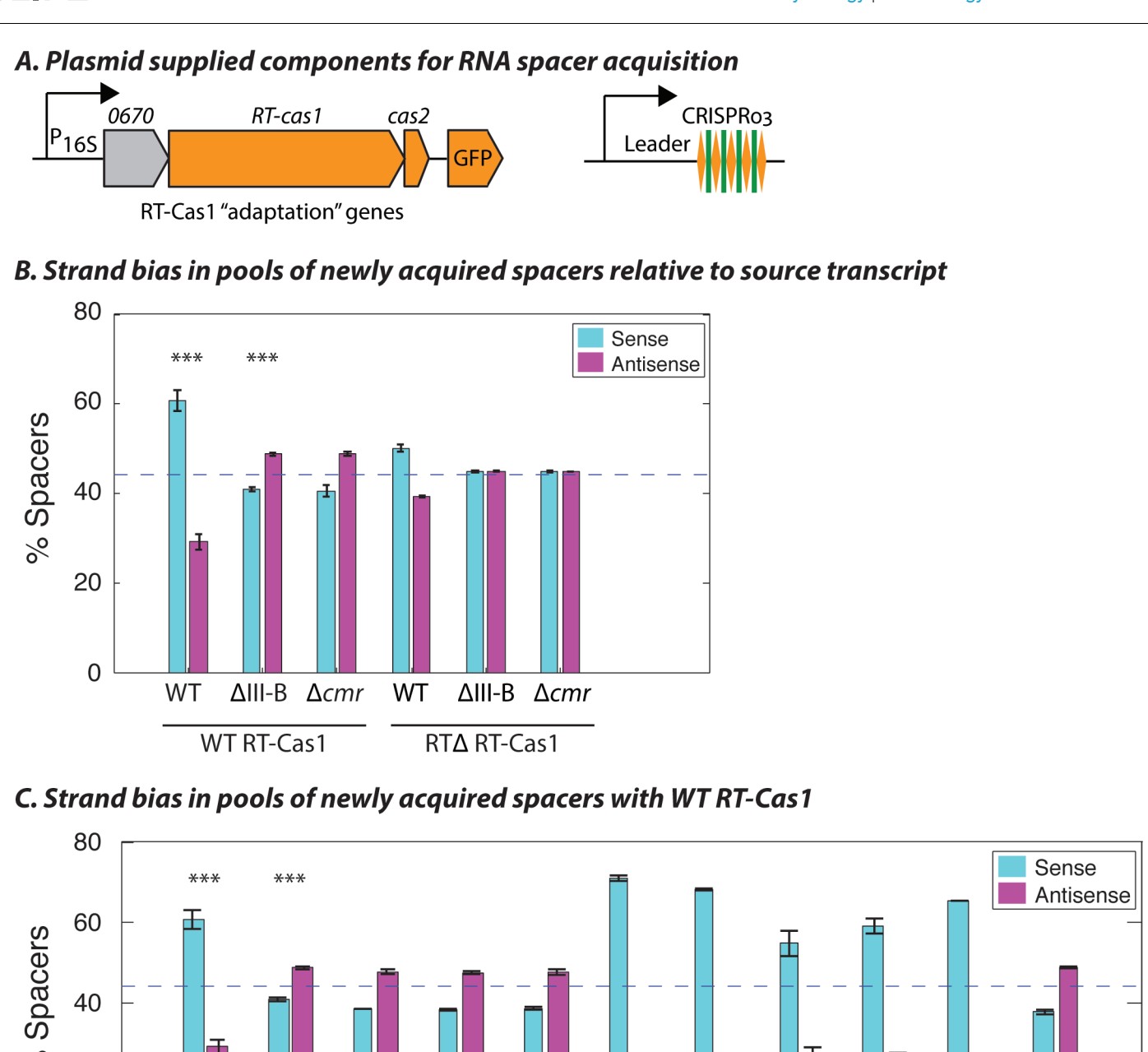

**Figure 2.** Interference-dependent strand bias in newly acquired spacers. (A) Arrangement of the type III-B CRISPR adaptation genes encoding Marme_0670, RT-Cas1, and Cas2 on a pKT230 broad-host-range vector under control of the putative 16S rRNA promoter (Adapted from [*Silas et al., 2016*]). (B) Proportion of newly acquired spacers isolated from CRISPR03 mapping to sense and antisense strands of host genes. Strains with the entire type III-B CRISPR-Cas operon deleted (ΔIII-B), the six *cmr* genes deleted (Δ*cmr*), or wildtype (WT) were tested with overexpression of WT or RT-deleted (RTΔ) versions of RT-Cas1. (C) Strand bias measurements of new spacers in additional mutants with overexpression of WT RT-Cas1. Data for WT and ΔIII-B strains redrawn from (B) for comparison. (B–C) As the WT-to-ΔIII-B comparison was most critical, these two measurements were repeated in eight samples for each genotype, with bars for standard error of the mean (SEM) shown. Other measurements were repeated twice, with bars indicating

*Figure 2 continued on next page*

*Figure 2 continued*

range. Blue dashed lines denote the expectation of no bias (no bias would yield ~44.3% [not 50%] because spacers mapping to regions outside of annotated genes [~11.4% of the total genome] provide us with no basis for strand assignment). Asterisks denote statistical testing was performed for the WT and ΔIII-B datasets (p<0.001; T-test).

DOI: https://doi.org/10.7554/eLife.27601.005

The following figure supplements are available for figure 2:

**Figure supplement 1.** Spacer acquisition in the absence of type III-B interference.

DOI: https://doi.org/10.7554/eLife.27601.006

**Figure supplement 2.** CRISPR spacer acquisition in *M. mediterranea* mutants.

DOI: https://doi.org/10.7554/eLife.27601.007

most closely related species of the same genus (*M. balearica* IVIA-Po-101; (*Espinosa et al., 2010*)), as well as other marine bacteria (*Pseudoalteromonas tunicata* and *Vibrio harveyi*) are resistant, indicating a narrow host range.

CPG1g infection of the ΔIII-B mutant was ~10 fold more productive than in WT, with substantially larger plaques in the ΔIII-B deletion (*Figure 3B,D*). This suggested a role of the type III-B CRISPR-Cas system in defense against CPG1g infection. Therefore, we scanned the phage genome for sequences matching spacers in each of the *M. mediterranea* MMB-1 CRISPR arrays. Surprisingly, none of the type III-B CRISPR spacers matched the phage genome, but the first two spacers in the type I-F CRISPR04 array had perfect or near-perfect matches to CPG1g (*Figure 3C*); the mismatch at position 6 for the first spacer is not expected to impair targeting efficiency in type I-F systems (*Vercoe et al., 2013*; *Cady et al., 2012*). Both near-perfect CPG1g-targeting spacers mapped to the antisense strand of predicted viral ORFs, indicating that they may be compatible with type III-B transcription-dependent interference (*Figure 3C*). In addition, the third type I-F CRISPR04 spacer weakly matched the phage genome (5 mismatches, including one in the 'seed' region previously shown to be critical for binding of the type I-F interference complex [*Wiedenheft et al., 2011*]) and had a sense (instead of antisense) orientation relative to the putative phage transcripts.

To test the role of the two CRISPR-Cas systems in defense against CPG1g, we measured the efficiency of plaquing (EOP) on strains lacking the type I-F and type III-B loci (ΔI-F, ΔIII-B, and ΔI-FΔIII-B). All three strains displayed an increase in phage sensitivity (*Figure 3D*), suggesting that there might be interplay between the two systems. Therefore, we removed the type I-F *cas1* and *cas2-3* genes without removal of the type I-F CRISPR04 array or *csy* genes (ΔI-F Cas123). This mutant strain was not impaired in defense against CPG1g, demonstrating that phage resistance was not dependent on either the effector nuclease or spacer acquisition machinery of the type I-F system. These observations suggest cooperation between the interference machinery of the type III-B system and the CRISPR array of the type I-F system in providing defense against CPG1g infection.

## Cross-talk between the type III-B system and the type I-F CRISPR array

As an independent method of assessing type I-F spacer-directed targeting by the type III-B system, we used the plasmid interference assay described earlier, except with target sequences derived from the portions of the CPG1g genome that matched the type I-F spacers 1 or 2. Consistent with transcription-dependent type III-B function, rather than type I-F DNA targeting, we observed interference only when the reverse complement of the spacer-matching sequence was transcribed (*Figure 3E*; also see *Figure 3—source data 1*). Moreover, we did not observe transcription-dependent interference when either the type III-B targeting module was disrupted or the type I-F CRISPR04 array was removed. Flanking the target sequence with type I-F CRISPR repeat sequences abrogated plasmid interference. Thus, despite the sparse similarity between the type I-F and type III-B repeat sequences (*Figure 1A*), the self/non-self discrimination mechanism of the type III-B system was still able to function. Confirming our finding that the type I-F effector nuclease (Cas3) was not required for resistance to CPG1g, deleting the type I-F *cas1* and *cas2-3* genes did not abolish transcription-dependent plasmid elimination (*Figure 3E*).

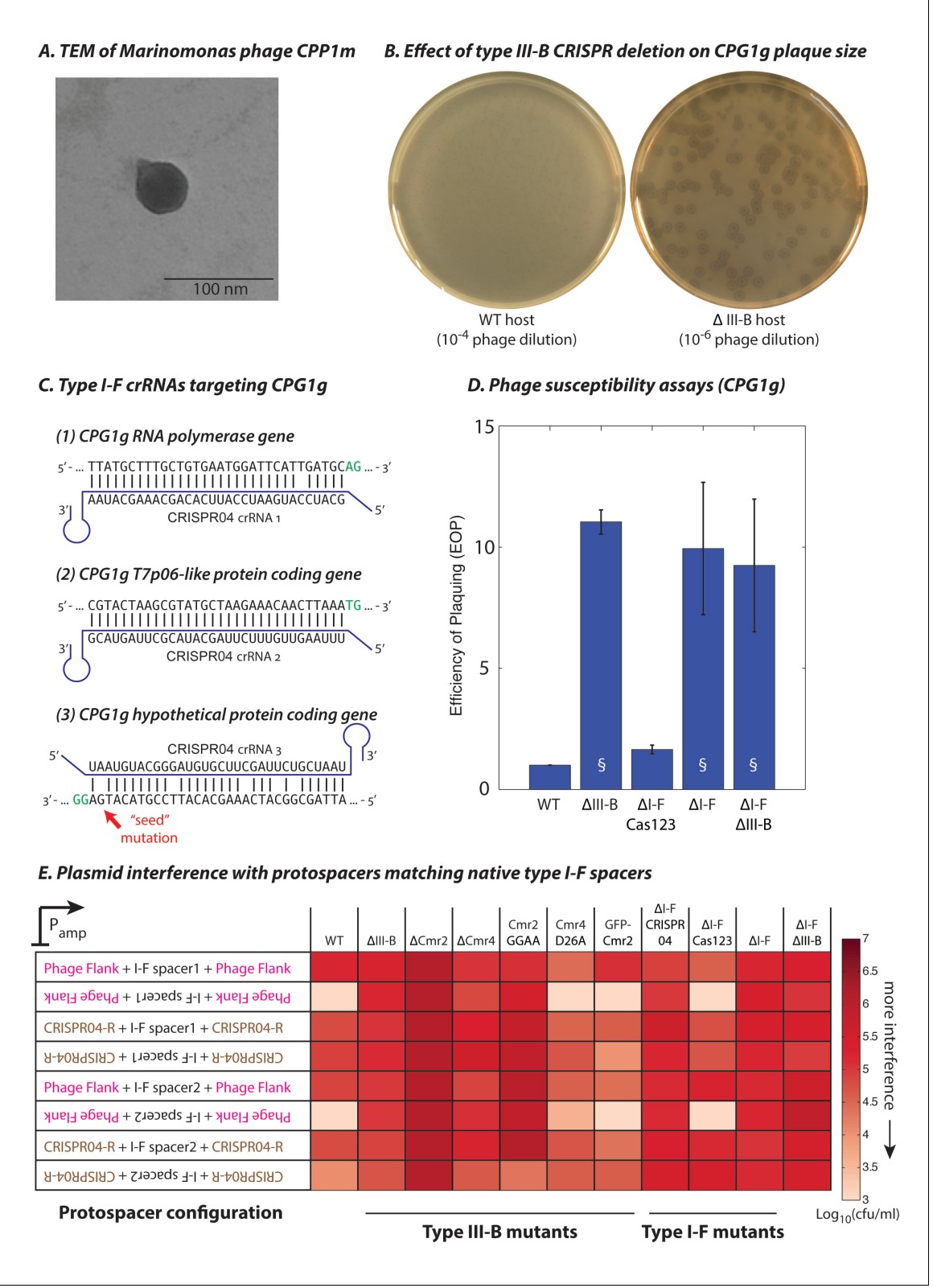

**Figure 3.** Phage-derived type I-F spacers are utilized by type III-B interference machinery. (A) Transmission electron micrograph (TEM) of phage CPP1m stained with uranyl acetate. (B) CPG1g plaques on lawns of WT (left) and ΔIII-B (right) strains. (C) The type I-F CRISPR04 array contains spacers with near-perfect 32 bp matches to portions of the CPG1g RNA polymerase and the T7p06-like genes, and a third spacer with 5 mismatches (including mutation in the 'seed' region) to a target in a gene encoding a hypothetical protein. CPG1g bases at the positions of the canonical type I-F PAM are

*Figure 3 continued on next page*

*Figure 3 continued*

highlighted in green. crRNAs from both near-perfect spacers 1 and 2 would be complementary to the putative phage mRNAs, while the mismatched 3rd spacer would yield crRNAs unable to bind targeted phage mRNA (denoted by inverted alignment). (D) Susceptibility of *M. mediterranea* mutants to CPG1g. Efficiency of plaquing is calculated as the fold change in counts of plaque forming units (PFU) relative to WT. Bars indicate results from 2 independent trials. § denotes enlarged plaques. (E) Transcription-dependent plasmid interference in various *M. mediterranea* mutants using native CRISPR04 spacers. Log-transformed cfu/ml measurements from conjugations of plasmids with various protospacer configurations. Type I-F spacer-matching sequences are flanked either by 28 bp of phage-derived sequence (pink text), or by 28 bp type I-F CRISPR repeats (brown text; CRISPR04-R). Upside down text denotes the reverse-complement.

DOI: https://doi.org/10.7554/eLife.27601.008

The following source data and figure supplement are available for figure 3:

**Source data 1.** Colony forming units per mL (cfu/ml) counts obtained from conjugations for plasmid interference assays.

DOI: https://doi.org/10.7554/eLife.27601.010

**Figure supplement 1.** Phylogenetic relationships of phage DNA polymerases.

DOI: https://doi.org/10.7554/eLife.27601.009

## The type III-B system uses pre-processed crRNAs from the type I-F system

Initially, we hypothesized that the type III-B system was capable of processing the type I-F pre-crRNA transcribed from the CRISPR04 array (*Figure 4A*). The genes encoding the type III-B pre-crRNA processing activity in *M. mediterranea* have not previously been identified, and the operon lacks a clear *cas6* homologue. Therefore, we tested whether the processing activity resided in the CRISPR adaptation components (RT-Cas1, Cas2, Marme_0670), as it was unlikely to reside in the six Cmr genes, and the only other gene in the operon – *marme_0671* – could not be easily tested because its overexpression led to toxicity. We expressed the adaptation components in the ΔIII-B and ΔI-FΔIII-B strains, and assayed for crRNA processing by small-RNA sequencing. Controls with only the CRISPR03 array supplied were also included. Pre-crRNA from the type III-B CRISPR03 array was processed efficiently in the ΔIII-B strain only when the type III-B adaptation components were supplied (*Figure 4B–E*). Thus, the factor(s) required for type III-B pre-crRNA processing do indeed reside in the supplied CRISPR adaptation module. Moreover, this demonstrates that the chromosomally-encoded type I-F factors (left intact in the ΔIII-B strain) could not process type III-B CRISPR03 pre-crRNA. In contrast, type I-F pre-crRNA was processed efficiently in the ΔIII-B strain, irrespective of the presence of the plasmid-encoded type III-B adaptation genes (*Figure 4F,G*).

The type I-F CRISPR04 crRNA signal is expected to be absent in the ΔI-FΔIII-B strain, because the type I-F operon deletion removed the entire CRISPR array (CRISPR04) and the pre-crRNA-processing factor Cas6f (Csy4) (*Figure 4H*). However, processed crRNAs from the type I-F orphan CRISPR05 array (see *Figure 1A*) were readily observed in the ΔIII-B strain, but not in the ΔI-FΔIII-B strain, regardless of whether type III-B adaptation genes were supplied (*Figure 4I*; also see *Figure 4—figure supplement 1*). These findings show that in the absence of Cas6f (Csy4), the type III-B processing machinery is not capable of generating mature crRNAs from type I-F CRISPR arrays. Thus, the processing machineries of both CRISPR-Cas systems are independent and specific to their respective CRISPR repeats. Taken together with the earlier results, this shows that the type III-B interference complex promiscuously obtains mature type I-F crRNAs, and can use them for transcription-dependent interference.

## PAM evasion allows escape from type I-F but not type III-B systems

The failure of the type I-F system to use its pre-existing spacers to combat CPG1g infection could be due to silencing of the type I-F system by phage- or host-encoded anti-CRISPRs, chemical modification of phage DNA, mutational escape by the phage, or atrophy of the type I-F system. Type I-F systems typically use GG PAMs (*Mojica et al., 2009*; *Rollins et al., 2015*), whereas the PAMs present in CPG1g for the targets of CRISPR04 spacer 1 and spacer 2 are AG and TG, respectively (*Figure 3C*). To test whether the *M. mediterranea* type I-F system was active and uses canonical GG PAMs, we repeated our plasmid interference assay using the phage-derived target sequences, except with the putative GG PAM restored (*Figure 5A,B*). In WT, we saw robust interference irrespective of the orientation of the target relative to the promoter (*Figure 5C*; also see *Figure 5—source data 1*); this was consistent with type I-F targeting of the plasmid DNA. Furthermore, deletion of the type III-B

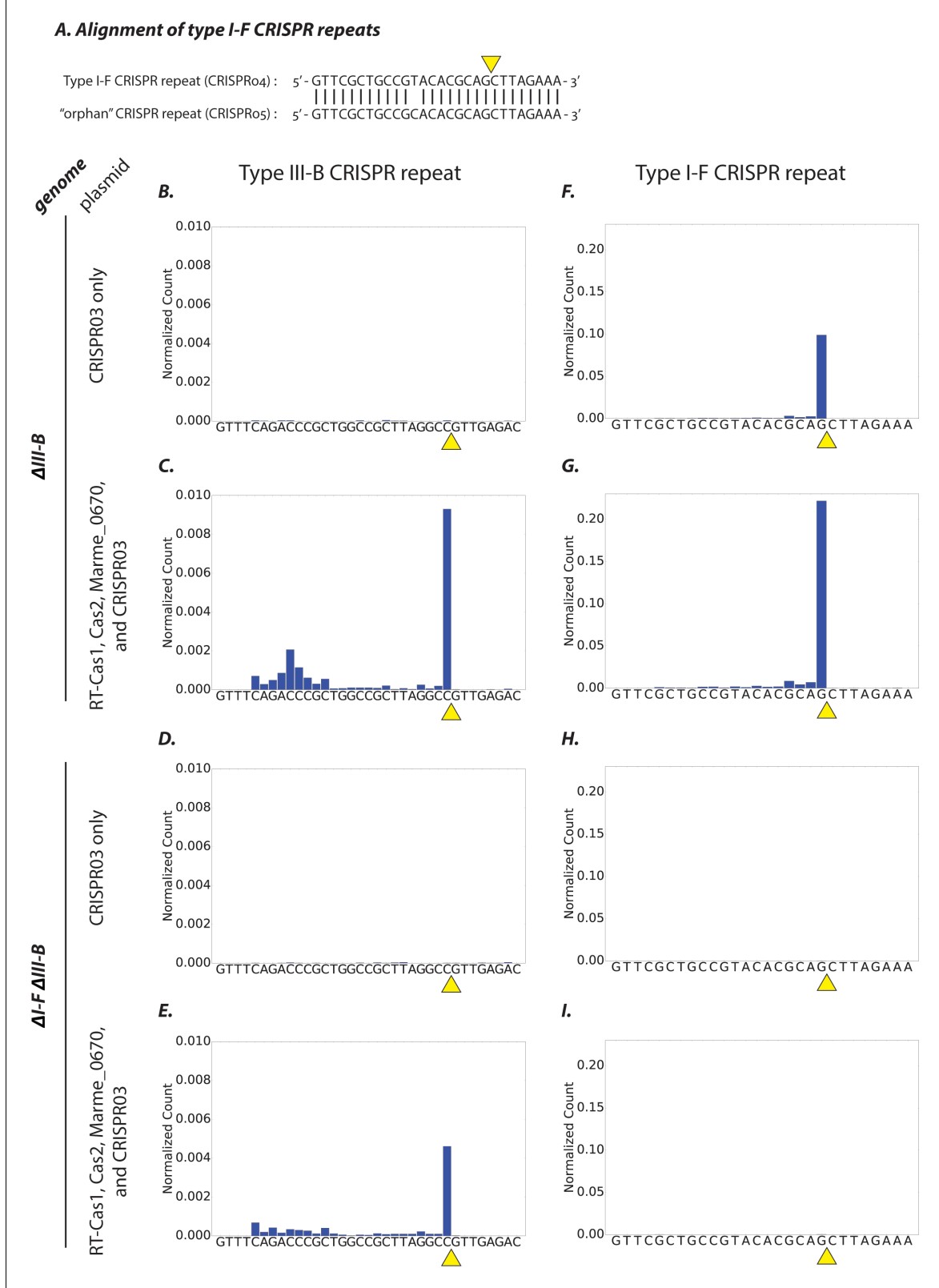

**Figure 4.** Type I-F and type III-B pre-crRNA processing in *M. mediterranea*. (**A**) Alignment of type I-F CRISPR repeat sequences from CRISPR04 and CRISPR05 (see *Figure 1A*). Yellow triangle marks the expected pre-crRNA processing site 8 nt upstream of 3' end of the CRISPR04/05 direct repeat sequence. (**B–I**) Processed crRNA levels assayed by high throughput small RNA sequencing. Each dataset includes RNA sequences from two separate transconjugants. The presence of a distinct 3' end sequence in the population of CRISPR repeat containing RNAs indicates site-specific cleavage and

*Figure 4 continued on next page*

*Figure 4 continued*

processing of pre-crRNA. Counts are normalized to Isoleucine tRNA levels (consistently the most abundant species encountered). Type III-B adaptation components – RT-Cas1, Cas2, Marme_0670 genes and the CRISPR03 array – were supplied in (**C, E, G, I**), whereas only the CRISPR03 array was supplied in (**B, D, F, H**). The genetic background of the host strain is indicated in the figure. The CRISPR04 repeat sequence is shown in (**F–I**) for simplicity, but read counts from both type I-F CRISPR arrays are included in the figures.

DOI: https://doi.org/10.7554/eLife.27601.011

The following figure supplement is available for figure 4:

**Figure supplement 1.** Type I-F pre-crRNA from the orphan array is not processed in the absence of type I-F factors.

DOI: https://doi.org/10.7554/eLife.27601.012

CRISPR-Cas operon did not impair plasmid elimination, whereas removal of the type I-F operon, in isolation or conjunction with the type III-B operon deletion, abrogated interference. Deletion of the type I-F *cas1* and *cas2-3* genes resulted in interference only when the reverse complement of the spacer sequence was transcribed from the plasmid. This indicates that type III-B targeting using type I-F crRNAs occurs concurrently with the type I-F targeting, but is only evident when type I-F interference is disabled. We also examined whether the type I-F targeting module could use type III-B crRNAs, by inserting GG PAM sequences in our plasmids containing type III-B protospacers. In this case, we observed only transcription-dependent plasmid elimination that is characteristic of type III interference (*Figure 5D*; also see *Figure 5—source data 1*). While it is possible that, due to some unforeseen protective mechanisms, we failed to see transcription-independent interference in this assay, these results suggest that the type I-F targeting module is unable to reciprocally use type III-B crRNAs for DNA interference.

By confirming that the type I-F system is active and relies on GG PAMs for target recognition, we conclude that the PAM mutations for the targets of spacer 1 and spacer 2 in phage CPG1g allow it to evade the type I-F system, and that the type III-B system is able to compensate for this. It is conceivable that sharing of spacers and specificities between CRISPR-Cas systems would result in redundancy by allowing type I spacers to be backed up in efficacy through the activity of co-occurring type III machinery. To evaluate the possible generality of such a backup process, we carried out experiments to test for spacer sharing in a substantially divergent host, *Serratia* sp. ATCC39006, that likewise carries both type I and type III CRISPR-Cas systems (*Fineran et al., 2013*; *Patterson et al., 2016*). Each CRISPR-Cas system in *Serratia* (types I-E, I-F and III-A) possesses at least one type-specific CRISPR array and spacer acquisition module. The type III-A operon encodes Cas1 and Cas2 but no RT-Cas1 fusion. To test for cross-type crRNA utilization in *Serratia*, we used a conjugation-based plasmid interference assay with targets matching native CRISPR spacers under the control of an arabinose-inducible promoter (*Figure 5—figure supplement 1A–D*). A control plasmid lacking any protospacer was not targeted, whereas a plasmid containing a type III target on the transcribed strand was subject to robust interference (*Figure 5—figure supplement 1E*). However, the type I-E and type I-F target-containing plasmids were only subject to interference when they possessed canonical PAMs for their respective systems (*Figure 5—figure supplement 1E*). Thus, the *Serratia* type III-A system does not appear to be capable of utilizing crRNAs from either the type I-E or type I-F systems.

## Dynamic interchange of type III-B CRISPR-Cas loci

We postulated that *M. mediterranea* MMB-1 once possessed type I-F-mediated immunity against the putative ancestor of phage CPG1g, and that subsequently a phage variant (such as CPG1g) with mutations in the target PAM sequences escaped the type I-F defenses. This would have led to selective advantage following acquisition of a PAM-independent CRISPR-Cas system (i.e. the type III-B operon) that was able to provide immunity against CPG1g by using the pre-existing phage-matching type I-F crRNAs. Consistent with this hypothesis, the type III-B operon in *M. mediterranea* MMB-1 is flanked by direct repeats reminiscent of integration scars (*Figure 1A*), suggesting that it was derived from a horizontal gene transfer event. To explore the putative mobility of this CRISPR-Cas system, we sequenced the genomes of two additional *M. mediterranea* strains – CPR1 isolated in Cabo de Palos (same site as the source of CPG1g and close to the site of isolation of *M. mediterranea* MMB-1), and IVIA-Po-186 isolated in Porto Colom in the Balearic Islands (~500 km away). All three strains shared >99.7% identity in 16S ribosomal RNA sequences, and >99% Average Nucleotide Identity

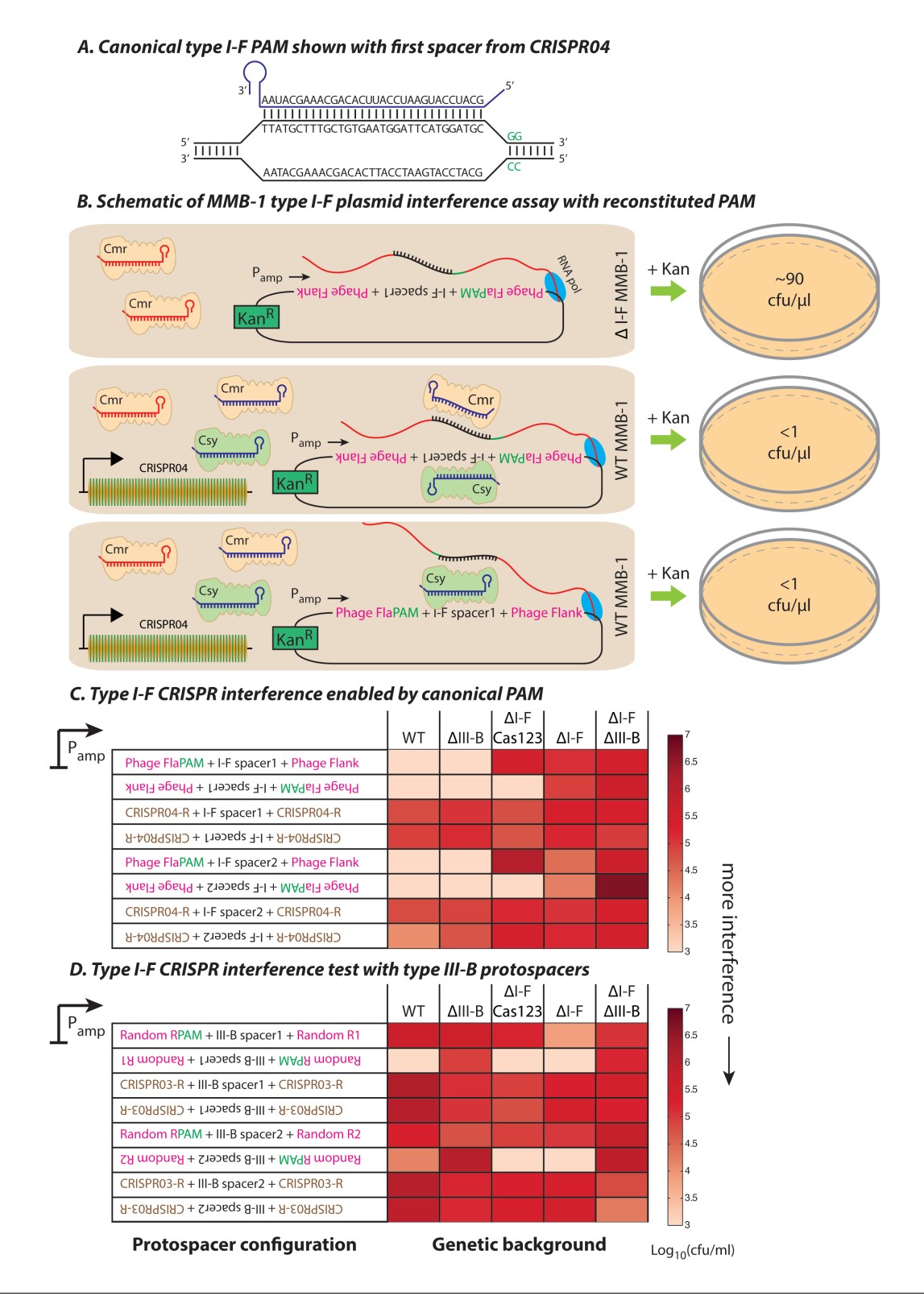

**Figure 5.** Contrasting determinants for self/non-self discrimination with a single crRNA. (**A**) Schematic of crRNA corresponding to the first spacer of CRISPR04 bound to protospacer DNA containing the canonical type I-F GG PAM. (**B**) Experimental outline of the plasmid interference assay using native CRISPR04 spacer sequences with reconstituted canonical (GG) PAMs. Both orientations of the protospacer DNA would be targeted by Csy-bound type I-F crRNA (blue), leading to plasmid loss independent of the direction of transcription. A configuration that would allow the protospacer

*Figure 5 continued on next page*

Figure 5 continued

RNA to be targeted by the Cmr-bound type I-F crRNA would also lead to plasmid instability. Conversely, deletion of the type I-F CRISPR-Cas system removes the CRISPR04 array, and would result in viable transconjugants despite the presence of Cmr-bound type III-B crRNA (red). Numbers shown correspond to cfu/μl measurements from the conjugations depicted. (C–D) Log-transformed cfu/ml measurements from conjugations of plasmids with different protospacer configurations. Upside down text denotes the reverse-complement. (C) Plasmid interference using native CRISPR04 spacers and GG PAM. The type I-F spacer-matching sequences are flanked either by 28 bp phage derived sequence (pink text) with reconstituted GG PAMs (green text), or by 28 bp type I-F CRISPR repeats (brown text; CRISPR04-R). (D) Plasmid interference using native CRISPR03 spacers and GG PAM. Type III-B spacer-matching sequences are flanked by Random R1 and R2 sequences (pink text) from *Figure 2C* with two bases preceding the protospacer converted to GG PAM (green text), or by the type III-B CRISPR repeat (brown text; CRISPR03-R). Data for CRISPR04-R and CRISPR03-R controls redrawn from *Figures 3E* and *2C*, respectively.

DOI: https://doi.org/10.7554/eLife.27601.013

The following source data and figure supplement are available for figure 5:

**Source data 1.** Colony forming units per mL (cfu/ml) counts obtained from conjugations for plasmid interference assays.

DOI: https://doi.org/10.7554/eLife.27601.015

**Figure supplement 1.** The *Serratia* type III-A system does not detectably utilize crRNAs from either of its type I-E or type I-F systems.

DOI: https://doi.org/10.7554/eLife.27601.014

(ANI) across their entire genomes, indicating that these are virtually identical strains of the same species, primarily differing in the accessory genome elements (including transposons and prophages) (*Kim et al., 2014*). *M. mediterranea* CPR1 and IVIA-Po-186 possess type I-F CRISPR-Cas systems that are highly similar to MMB-1 (*Figure 6A*), except that none of their spacers match the CPG1g genome. This is consistent with their higher sensitivity (than MMB-1) to this virus. Although a few identical spacers were found between the CPR1 and IVIA-Po-186 strains, the overall lack of conservation of the spacer repertoire between even the cognate type I-F CRISPR arrays suggests a high diversity of invasive genetic elements in the native environments of these hosts.

Moreover, IVIA-Po-186 and CPR1 do not possess the MMB-1 type III-B operon. The CPR1 genome contains a highly divergent type III-B CRISPR-Cas system in a different genomic location to the MMB-1 type III-B operon (*Figure 6B*). Moreover, the repeats of the MMB1 and CPR1 type III-B CRISPR arrays are dissimilar (~50% identity). These data are indicative of the independent horizontal acquisition of a distinct type III-B system in this strain. The IVIA-Po-186 genome contains a putatively defunct CRISPR array with degraded type III-B repeats, which is present within the same genomic context as the type III-B locus in MMB-1. This indicates that a type III-B system likely once resided in IVIA-Po-186 (*Figure 6B*). We hypothesize that once selective pressure for maintaining the supposed type III-B system alleviated (for example the threat of a specific phage abated), the protein-coding operon was lost due to recombination between flanking CRISPR arrays. No remnant of the MMB-1 type III-B system was found in CPR1, and no remnant of the CPR1 system was found in either MMB1 or IVIA-Po-186. Overall, our observations are in agreement with the notion that the type III-B systems in these closely related *M. mediterranea* strains were independently acquired by horizontal transfer.

Our findings support a model in which specific type III CRISPR-Cas systems can serve as promiscuous 'backup units' for type I-F adaptive immunity in *Marinomonas* species. We also analyzed the CRISPR-Cas content of previously sequenced *Marinomonas* genomes and found that *Marinomonas* sp. MWYL1 possesses a chimeric CRISPR-Cas system containing a type III-B interference module adjacent to a gene encoding the type I-F pre-crRNA processing enzyme Cas6f (Csy4) and a type I-F CRISPR array (but no type III-B Cas6 or CRISPR array; *Figure 6C*). There is also a type I-E system but no type III CRISPR arrays in this strain. The closest homolog of the MWYL1 Cas6f protein belongs to a bonafide type I-F operon in *Marinomonas gallaica* (*Figure 6C*). By searching for co-occurring *cas10* and *cas6f* genes, we identified a similar hybrid in *Shewanella putrefaciens* 200 (NC_017566). This suggests that cross-talk between type III-B and type I-F systems is not limited to *Marinomonas*. We propose that selective pressures resulting from proliferation of phage mutants that evade host type I-F defenses can drive the selection for horizontally acquired type III-B systems, and may have also resulted in selection for chimeric type III-B/Cas6f type I-F CRISPR loci in some hosts.

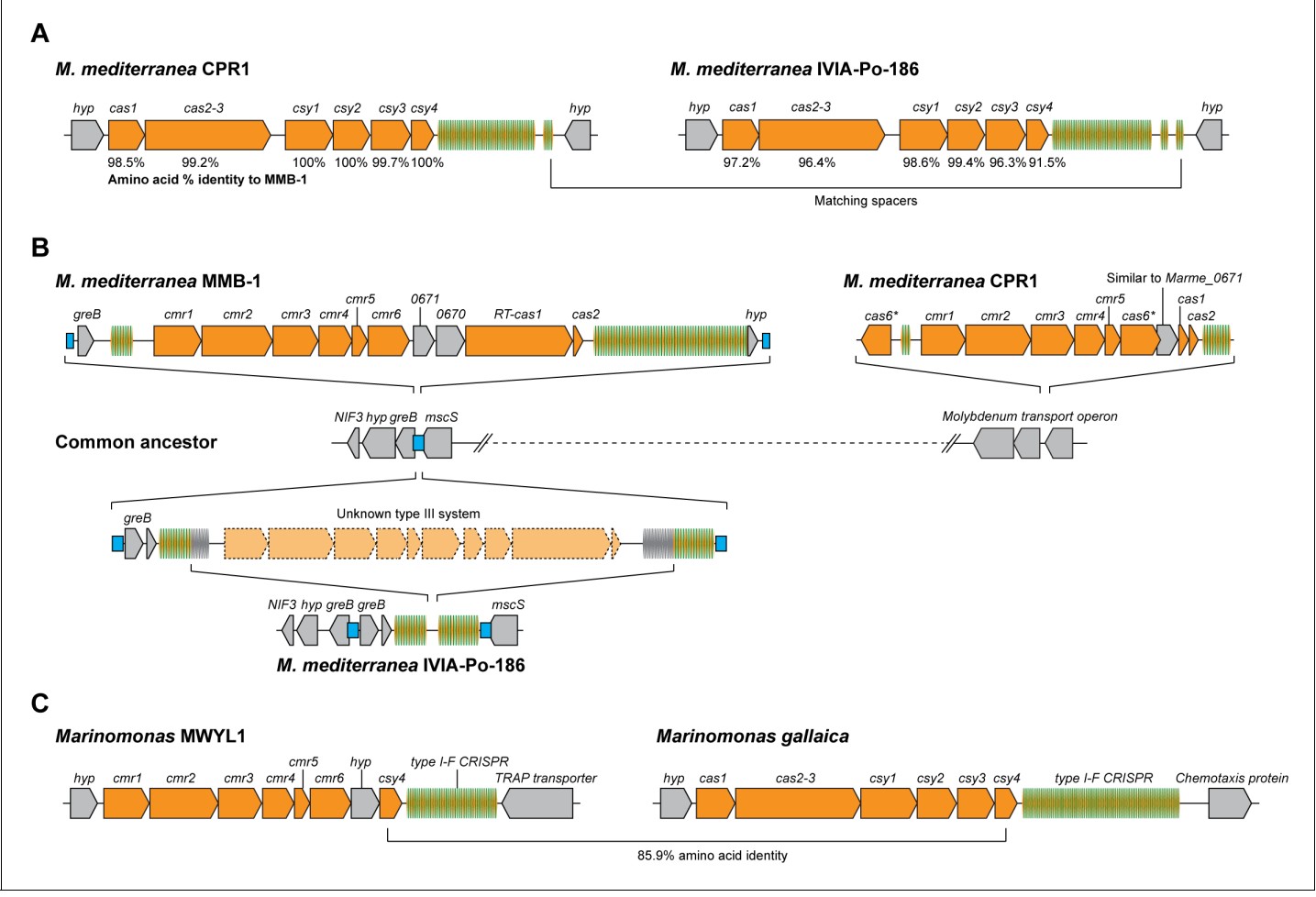

**Figure 6.** Arrangements of type I-F and type III-B CRISPR-Cas systems in *Marinomonas*. (**A**) Schematic of the type I-F CRISPR-Cas loci in *M. mediterranea* CPR1 and IVIA-Po-186. The numbers below genes indicate % protein sequence identity to corresponding homologs in *M. mediterranea* MMB-1. (**B**) Schematic of the type III-B loci in the *M. mediterranea* strains that we sequenced. A hypothetical common ancestor lacking any type III-B systems is shown. The *M. mediterranea* MMB-1 type III-B system is integrated between the *greB* and *mscS* genes. The same genomic locus contains a degraded CRISPR array in *M. mediterranea* IVIA-Po-186, possibly as a result of recombination between CRISPRs of an ancestral type III-B operon at this site. A different type III-B system is found in *M. mediterranea* CPR1, integrated within the molybdenum transport operon. The *greB-mscS* regions in *M. mediterranea* IVIA-Po-186 and CPR1 were confirmed by PCR and sanger sequencing. (**C**) The *Marinomonas* sp. MWYL1 genome contains a chimeric CRISPR-Cas system with *cmr* genes encoding the type III-B effector complex and the type I-F pre-crRNA processing enzyme Cas6f, along with a type I-F CRISPR array. The most similar homolog of the *Marinomonas* sp. MWYL1 Cas6f (Csy4) is encoded in the *M. gallaica* type I-F system.
DOI: https://doi.org/10.7554/eLife.27601.016

## The type III-B CRISPR-Cas system counteracts viral escape from the type I-F system

We hypothesized that it might be possible to detect phages in the native ecosystem that remain sensitive to *M. mediterranea* MMB-1 type I-F interference. Therefore, we performed additional phage isolations using samples of sediment obtained from different *P. oceanica* meadows off the Mediterranean coast, with the host strain *M. mediterranea* T103 (*Lucas-Elío et al., 2002*) that showed high sensitivity to CPG1g in preliminary assays. Another new phage, named CB5A, was isolated and genome sequencing revealed that CB5A is a close relative of phages CPG1g/CPP1m, with only a few differences (*Figure 7—source data 1*). Most notably, the protospacer targeted by the second spacer in the *M. mediterranea* MMB-1 type I-F CRISPR04 array contained a canonical PAM (GG) (*Figure 7A*). Moreover, the protospacer in phage CB5A that matches spacer 3 in the host CRISPR04 array contains only 3 mismatches versus 5 for phage CPG1g. Therefore, we propose that CB5A more closely resembles the 'pre-escape' ancestor of CPG1g. CB5A showed almost no ability

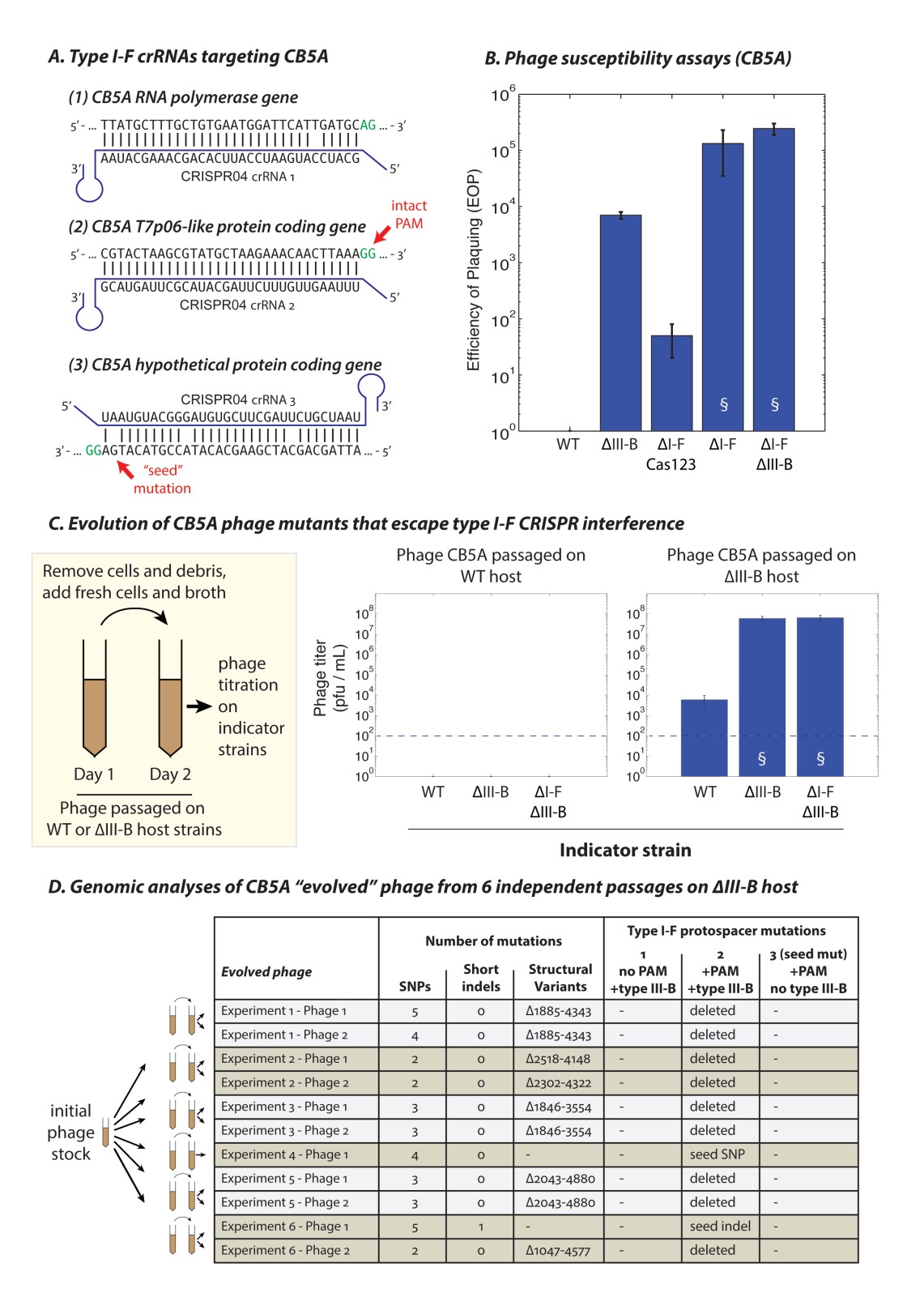

**A. Type I-F crRNAs targeting CB5A**

*(1) CB5A RNA polymerase gene*

5' - ... TTATGCTTTGCTGTGAATGGATTCATTGATGCAG ... - 3'

3'| AAUACGAAACGACACUUACCUAAGUACCUACG |5'

CRISPR04 crRNA 1

*(2) CB5A T7p06-like protein coding gene*

intact PAM

5' - ... CGTACTAAGCGTATGCTAAGAAACAACTTAAAGG ... - 3'

3'| GCAUGAUUCGCAUACGAUUCUUUGUUGAAUUU |5'

CRISPR04 crRNA 2

*(3) CB5A hypothetical protein coding gene*

CRISPR04 crRNA 3

5' UAAUGUACGGGAUGUGCUUCGAUUCUGCUAAU |3'

3' - ... GGAGTACATGCCATACACGAAGCTACGACGATTA ... - 5'

"seed" mutation

**B. Phage susceptibility assays (CB5A)**

Efficiency of Plaquing (EOP) vs strains: WT, ΔIII-B, ΔI-F Cas123, ΔI-F, ΔI-F ΔIII-B

**C. Evolution of CB5A phage mutants that escape type I-F CRISPR interference**

Remove cells and debris, add fresh cells and broth

Day 1 Day 2

Phage passaged on WT or ΔIII-B host strains

phage titration on indicator strains

Phage CB5A passaged on WT host

Phage CB5A passaged on ΔIII-B host

Phage titer (pfu / mL)

Indicator strain: WT, ΔIII-B, ΔI-F ΔIII-B

**D. Genomic analyses of CB5A "evolved" phage from 6 independent passages on ΔIII-B host**

initial phage stock

| Evolved phage | Number of mutations | | | Type I-F protospacer mutations | | |
|---|---|---|---|---|---|---|
| | SNPs | Short indels | Structural Variants | 1 no PAM +type III-B | 2 +PAM +type III-B | 3 (seed mut) +PAM no type III-B |
| Experiment 1 - Phage 1 | 5 | 0 | Δ1885-4343 | - | deleted | - |
| Experiment 1 - Phage 2 | 4 | 0 | Δ1885-4343 | - | deleted | - |
| Experiment 2 - Phage 1 | 2 | 0 | Δ2518-4148 | - | deleted | - |
| Experiment 2 - Phage 2 | 2 | 0 | Δ2302-4322 | - | deleted | - |
| Experiment 3 - Phage 1 | 3 | 0 | Δ1846-3554 | - | deleted | - |
| Experiment 3 - Phage 2 | 3 | 0 | Δ1846-3554 | - | deleted | - |
| Experiment 4 - Phage 1 | 4 | 0 | - | - | seed SNP | - |
| Experiment 5 - Phage 1 | 3 | 0 | Δ2043-4880 | - | deleted | - |
| Experiment 5 - Phage 2 | 3 | 0 | Δ2043-4880 | - | deleted | - |
| Experiment 6 - Phage 1 | 5 | 1 | - | - | seed indel | - |
| Experiment 6 - Phage 2 | 2 | 0 | Δ1047-4577 | - | deleted | - |

**Figure 7.** The type III-B system suppresses proliferation of phage mutants that escape type I-F defenses. (**A**) The type I-F CRISPR04 array contains spacers with near-perfect 32 bp matches to targets in the CB5A RNA polymerase and the T7p06-like genes, and a third spacer with 3 mismatches (including a mutation in the 'seed' region) to a target in a gene encoding a hypothetical protein. CB5A bases at the positions of the canonical type I-F PAM are highlighted in green. The intact type I-F PAM (GG) at the second protospacer site is highlighted. crRNAs from both near-perfect spacers 1

*Figure 7 continued on next page*

eLIFE Research article                    Genomics and Evolutionary Biology | Microbiology and Infectious Disease

Figure 7 continued

and 2 would be complementary to the putative phage mRNAs, while the mismatched 3rd spacer would yield crRNAs unable to bind targeted phage mRNA (denoted by inverted alignment). (B) Susceptibility of *M. mediterranea* mutants to CB5A. Efficiency of plaquing (EOP) is calculated as the fold change in counts of plaque forming units (PFU) relative to WT. CB5A titers on the WT host (defined here as EOP = 1) could only be obtained at phage concentrations near the assay detection limit (100 pfu/mL). Bars indicate results from 2 independent trials. § denotes enlarged plaques. (C) Cultures of the WT and ΔIII-B strains of *M. mediterranea* MMB-1 were inoculated with phage CB5A, and phage populations were sub-cultured the next day with fresh host cells. Phage titers were determined on WT, ΔIII-B, and ΔI-FΔIII-B indicator strains using the Most Probable Number method (*Kott, 1966*). Titers of phage populations passaged on WT and ΔIII-B strains are from 3 and 6 independent experiments, respectively. § denotes enlarged plaques. Asterisks denote statistically significant differences relative to the phage titers on the WT strain (p<0.001; T-test). Phage stocks generated by passaging on WT host did not show any infectivity, even when used undiluted. Assay detection limit is indicated by dashed blue lines. (D) Whole genome sequencing assessment of the mutational landscape of 11 isolates of 'escape' phage from 6 evolution experiments with the ΔIII-B host. The numbers of single nucleotide polymorphisms (SNPs), short insertions or deletions (indels), and large insertions, deletions, or rearrangements (structural variants) are shown. Precise chromosomal breakpoints of deletions (denoted by Δstart-end) are indicated. Corresponding mutations in the type I-F protospacer regions are also listed. Protospacers are numbered according to (A); the presence of a PAM, and the compatibility of the protospacer with type III-B interference (i.e. whether the type I-F crRNA is complementary to the phage mRNA) are indicated in the column headers for reference.

DOI: https://doi.org/10.7554/eLife.27601.017

The following source data and figure supplements are available for figure 7:

**Source data 1.** Percent Identity and Similarity comparisons of protein products of homologous genes from phages CPG1g and CB5A.
DOI: https://doi.org/10.7554/eLife.27601.020
**Figure supplement 1.** Plaque assays with 'original' and 'evolved' CB5A phage populations.
DOI: https://doi.org/10.7554/eLife.27601.018
**Figure supplement 2.** Plaque assays with 'evolved' CB5A phage isolates.
DOI: https://doi.org/10.7554/eLife.27601.019

to infect WT MMB-1, with extremely small plaques forming only at very high phage concentrations (near the assay limit of detection) (*Figure 7B*, also see *Figure 7—figure supplement 1*). In the absence of type I-F mediated defense (the ΔI-F Cas123 strain) we observed a ~50 fold increase in sensitivity to CB5A (*Figure 7B*), but with very small plaques (*Figure 7—figure supplement 1*). This demonstrates that interference by the type I-F system contributes to immunity against CB5A – consistent with the intact PAM for the protospacer matched by the *M. mediterranea* MMB-1 CRISPR04 spacer 2. This is in contrast to CPG1g infection (i.e. a PAM-escape mutant), where the type I-F system alone does not confer measureable immunity (*Figure 3D*). Moreover, deletion of the MMB-1 type III-B module resulted in a >1000 fold increase in sensitivity to CB5A (relative to WT host) (*Figure 7B*). Thus, the type III-B system provides greatly enhanced defense against CB5A infection. However, CB5A 'pre-escape phage' plaques on the ΔIII-B indicator strain were still smaller than those of CPG1g on the same indicator strain (compare *Figure 7—figure supplement 1* and *Figure 3B*), likely due to type I-F–mediated targeting of the PAM-containing protospacer in CB5A. Maximal sensitivity to CB5A, evident in both plaquing efficiency and plaque size, was observed when both the type I-F and type III-B systems were removed, or in the strain with the type I-F operon deleted – in the latter case this is a consequence of removal of the type I-F CRISPR array containing the phage-matching spacers. These findings demonstrate that both systems participate in coordinated defense against CB5A using the type I-F crRNAs.

To examine the conditions that led to proliferation of the CPG1g/CPP1m 'escape' phages and/or selection for the uptake and maintenance of the promiscuous type III-B system in *M. mediterranea* MMB-1, we performed an experiment to measure the frequency of phage escape in the presence or absence of type III-B immunity. Cultures of WT and ΔIII-B strains were infected with CB5A, and the phage populations were passaged on fresh host cells the following day. The resulting phage titers were then determined on several host mutants. We found that the phage populations had been eliminated from the WT cultures in just one re-inoculation step, with the absence of any detectable plaque formation on any indicator strains (*Figure 7C*). In contrast, phage populations passaged through cultures of the ΔIII-B strain yielded small plaques on lawns of the WT indicator strain, suggesting the presence of 'escape phage' (*Figure 7—figure supplement 1*). These phage populations displayed a ~ 5,000–10,000 fold increased EOP for infection of ΔIII-B versus WT indicator strains, and no additive effect compared with the ΔI-FΔIII-B strain (*Figure 7C*). Moreover, the plaque sizes were greatly increased on the ΔIII-B indicator strain, unlike the original CB5A phage that produced small plaques on ΔIII-B lawns (*Figure 7—figure supplement 1*). This indicates that passaging of

CB5A in cells possessing only type I-F immunity results in rapid proliferation of phage variants that evade the host type I-F defenses. To confirm genetic changes to the phage population, we attempted to sequence the genomes of 2 escape phages isolates from each of the 6 independent evolution experiments with the ΔIII-B host. All 11 successfully sequenced phage genomes contained a deletion or seed-region mutation in the PAM-containing protospacer (subject to type I-F interference), but not in the PAM-lacking protospacer, nor in the mismatched protospacer that already contained a seed mutation (*Figure 7D*). In all, 8 distinct escape mutations (6 large deletions spanning the entire protospacer, 1 seed substitution, 1 seed frameshift) were observed (*Figure 7D*). None of the escape mutations were observed across multiple independent trials of the evolution experiment, suggestive of a diversity of escape phages which could not proliferate when passaged with the WT host due to simultaneous PAM-independent type III-B targeting at both the first and second protospacers. Finally, we tested the ability of three escape phages (containing the seed mutation, the seed frameshift, and a deletion spanning the PAM-containing protospacer respectively) to infect various *M. mediterranea* mutants. We found small plaques on WT and ΔI-F Cas123 indicator strains, and large plaques on ΔIII-B and ΔI-FΔIII-B hosts for all three escape phages (*Figure 7—figure supplement 2*), similar to the naturally occurring escape phage CPG1g. These findings confirm that in the absence of the *M. mediterranea* MMB-1 type III-B CRISPR-Cas system, phage variants that escape host type I-F immunity, such as CPG1g, can proliferate. Thus, host strains that possess, or in theory acquire, type III-B systems that can utilize existing host type I-F crRNAs, have a fitness advantage when faced with rapidly evolving threats.

## Discussion

Recently, we showed that the type III-B RT-Cas1 fusion in *M. mediterranea* enables spacer acquisition directly from RNA (*Silas et al., 2016*). Here we demonstrated that this system directs DNA interference in a transcription-dependent manner. Complementarity between crRNA guides and target RNA is a requirement for this mechanism, hence new spacers (whether DNA or RNA derived) need to be integrated directionally in the CRISPR array to function. By eliminating biases associated with autoimmunity, we showed that the orientation of new spacers derived from host RNAs was only marginally in favor of the functional (antisense) direction. Therefore, almost half of the RT-Cas1 acquired spacers are useless for RNA and DNA interference. This is surprising and raises the question of why higher fidelity spacer orientation mechanisms, such as observed in PAM-dependent CRISPR-Cas systems (*Shmakov et al., 2014*; *Staals et al., 2016*), have not arisen.

Characterizing *M. mediterranea* defense against phages, we uncovered an ecologically functional plasticity in crRNA selection and utilization by the type III-B interference machinery that allows for cross-talk between the (otherwise autonomous) type III-B system and spacers stored in a type I-F CRISPR array. We show that the type III-B effector complex can use type I-F crRNAs despite the stark dissimilarity between the respective CRISPR repeat sequences. Use of type I-F crRNAs by the type III-B system provides protection to *M. mediterranea* MMB-1 from infection by a phage abundant in the native environment of the host. The type I-F system contains immunological memory of the phages isolated in this study, CPG1g/CPP1m and CB5A, but is unable to provide effective defense against CPG1g due to PAM mutations in the phage genome. Type III-B interference against CPG1g (and other phage variants that 'escape' type I-F defenses) using type I-F spacers, but without requiring a specific PAM, serves as an additional line of defense for *M. mediterranea*. We found that this cross-system spacer usage is reliant on the type I-F pre-crRNA processing activity of Cas6f (Csy4). Consistent with this, a chimeric CRISPR-Cas system in the related bacterium *Marinomonas* sp. MWYL1 comprises a set of genes encoding a type III-B effector complex, with a type I-F *cas6f* gene and type I-F CRISPR array. In addition, the ability to process type I pre-crRNAs might be carried within some type III operons; for example, the type III-B system in *M. mediterranea* CPR1 encodes two diverse Cas6-family proteins (*Figure 6*) that may enable the system to independently process pre-crRNAs from different CRISPR-Cas systems. This would be interesting for future study.

Although it is possible that the *M. mediterranea* MMB-1 type III-B system gained the ability to use type I-F crRNAs during prolonged co-residence with the type I-F system, our comparative genomic data support an alternate theory. We propose that CPG1g escaped the defenses of a progenitor *M. mediterranaea* host – which only contained a type I-F system – through the acquisition of PAM mutations. Proliferation of the escape phage drove selection for *M. mediterranea* strains that had

subsequently acquired the type III-B CRISPR-Cas system, which could co-opt the existing type I-F crRNAs to provide defense against CPG1g. Despite the fact that the *M. mediterranea* MMB-1 strain containing the CPG1g- and CB5A-matching spacers was isolated ~20 years ago (*Solano et al., 1997*), the phage protospacer sequences do not appear to have acquired further mutations to avoid type III-B interference. Previous studies have shown that type III systems are more tolerant of mutations in their target protospacer and flanking sequences (*Tamulaitis et al., 2014*; *Elmore et al., 2016*; *Marraffini and Sontheimer, 2010*; *Staals et al., 2014*). Thus, the probability of escape from the type III system is likely to be less than for type I-F systems. Additionally, the type III-B system might provide a protective advantage to *M. mediterranea* MMB-1 without significantly affecting the phage population, which might rely on other bacterial reservoirs (*Haydon et al., 2002*); for example, the CPR1 and IVIA-Po-186 *M. mediterranea* isolates are sensitive to CPG1g infection.

Such 'immunological insurance' provided by mobile CRISPR-Cas modules might be generalizable to other type III CRISPR-Cas systems. Although we did not detect cross-utilization of type I-E and type I-F spacers by the type III-A system in *Serratia*, this potentially reflects the evolutionary divergence of type III-A from type III-B/C/D CRISPR-Cas systems. Type III-A systems typically contain their own CRISPR adaptation and processing factors, which are often lacking in type III-B/C/D systems (*Makarova et al., 2015*). The diversity of type III-B CRISPR-Cas configurations co-occurring with type I systems in *Marinomonas* species, and the presence of the chimeric type III/I-F system in *Marinomonas* sp. MWYL1 suggest that the use of type I crRNAs by type III-B effector modules is not limited to *M. mediterranea* MMB-1. The ability to borrow crRNAs from disparate CRISPR loci could explain why type III-B systems also often lack CRISPR arrays (*Makarova et al., 2015*). For example, a type III-B system lacking CRISPR arrays in *Sulfolobus islandicus* REY15A has been reported to utilize crRNAs from a type I-A locus for interference in vivo (*Deng et al., 2013*). These type III systems might have lost their system-specific CRISPR arrays over time. Alternatively, we propose they could have been acquired by horizontal gene transfer as promiscuous 'backup' interference modules, adding an additional layer of support to the host type I systems. Consistent with this, genomic analyses of other *S. islandicus* strains revealed that while the type I-A locus is always present, modular reassortments of type III Cas modules are common (*Held et al., 2013*).

One hypothesis that might explain the difference between type III-A and other type III systems is that the effector complexes encoded by type III-B/C/D loci could possess a higher degree of plasticity in their crRNA selection criteria. Certainly, in *M. mediterranea* MMB-1, the type III-B effector complex can use its own type III-B associated arrays, but is also able to utilize information stored in the type I-F CRISPR array despite both loci possessing highly dissimilar repeat sequences. This plasticity could not be examined in previous studies (*Deng et al., 2013*; *Elmore et al., 2016*) because the host genomes in those studies only contained type I CRISPR arrays with identical repeats. Future research will be required to determine the molecular requirements that enable plasticity in crRNA usage by type III systems. For instance, previous studies have documented the processing and tight interaction between Cas6f and mature crRNAs (*Haurwitz et al., 2010*; *Przybilski et al., 2011*), suggesting the possibility of direct interactions between Cas6f and components of the type III-B Cmr complex. With well-characterized CRISPR adaptation and immunity pathways and an accessible habitat for the isolation of natural strains and parasites, *M. mediterranea* could serve as a valuable model system for interrogating such interactions. A high degree of plasticity in crRNA usage criteria would enhance the success of mobile genetic elements containing type III interference modules, as they could be beneficial immediately upon acquisition by a CRISPR-compatible host.

If widespread, plasticity in crRNA selection by type III interference modules could help to explain the frequent co-occurrence of type III and type I systems in prokaryotic genomes. In this regard, it is salient that we observed adherence of the type III self/non-self discrimination mechanism with type I-F crRNAs. This mechanism permits the 'universal' avoidance of self-targeting, without a specific sequence requirement such as a PAM, potentially allowing for broad non-specific crRNA use by type III-B systems. The emerging picture is one of fluid interactions between bacterial genomes and a diversity of mobile elements, including not only phage and plasmids but CRISPR-Cas systems as well (*Godde and Bickerton, 2006*). Our findings demonstrate that longitudinal studies of CRISPR adaptation in ecological contexts (*Andersson and Banfield, 2008*) could benefit from whole genome sequencing of highly-related host strains to identify horizontal transfer of complete CRISPR-Cas modules and de novo formation of chimeric systems.

## Materials and methods

### High throughput sequencing data

All sequencing data generated in this study (SRP103952) were deposited at the NCBI Short Read Archive (SRA) and are additionally summarized in *Supplementary file 1*.

### Plasmid construction

The plasmid constructs used for spacer acquisition and in vivo pre-crRNA processing studies in *M. mediterranea* were described previously (*Silas et al., 2016*). Plasmids for CRISPR interference assays were built on the pKT230 backbone (a gift from Prof. L. Banta, Williams College). Protospacer RNA transcription was driven by a constitutive beta-lactamase promoter ($P_{amp}$) in pKT230. By high-throughput RNA sequencing we previously confirmed that this promoter drives efficient unidirectional transcription in MMB-1 (data not shown). Plasmids for the *Serratia* interference assays were based on the pPF781 backbone (*Patterson et al., 2016*). Protospacer RNA transcription was driven by the arabinose-inducible $P_{BAD}$ promoter. All plasmids were verified by sequencing and are available upon request. Plasmid sequences are included in *Supplementary file 2*.

### Strains and culture conditions

Mutant *M. mediterranea* MMB-1 strains were constructed in a spontaneous Rifampicin-resistant genetic background by allelic exchange mutagenesis using *sacB*/sucrose counter-selection with the pEX18Gm suicide vector backbone and the *E. coli* S17-1λpir donor strain as described previously (*Campillo-Brocal et al., 2013*). The genomes of all mutant strains were verified by whole genome sequencing. Despite several attempts, we were unable to obtain strains with deletions of the *csy* genes.

All bacterial strains were stored at −80°C in 20% glycerol. Two clones (independent transconjugants) from each conjugation were tested for spacer acquisition and in vivo pre-crRNA processing assays. Plasmids were mobilized into *M. mediterranea* as previously described (*Solano et al., 2000*). Transconjugants were selected on 2216 marine agar (Difco) with 50 µg/mL Kanamycin and 50 µg/mL Rifampicin at 25°C. For nucleic acid extraction, transconjugants were inoculated in 2 mL Km-broth (2216 marine medium (Difco) with 50 µg/mL Kanamycin) and shaken at 23–25°C for 4–8 hr. Cultures were immediately expanded in 15 mL Km-broth and grown for an additional 4–8 hr before nucleic acid extraction.

### Nucleic acid extractions

This method is a slight modification of a previously published protocol for CRISPR spacer sequencing (*Silas et al., 2016*); we have provided the protocol in entirety for completeness, retaining relevant text from the original protocol. Total RNA for in vivo spacer processing assays was extracted from 300 to 500 µL of 15 mL confluent cultures using TRIZOL reagent (Life Technologies) according to manufacturer's instructions, without any subsequent enzymatic treatments. The remaining 14.5 mL of culture was used for plasmid midiprep. Cells were harvested by centrifugation (4000 x g, 30 min, 4°C) and homogenized in 300 µL alkaline lysis buffer (40 mM glucose, 10 mM Tris, 4 mM EDTA, 0.1 N NaOH, 0.5% SDS) at 50°C by vortexing until clear (10–15 min). Chilled neutralization buffer (600 µL of 3 M $CH_3COOK$, 2 M $CH_3COOH$) was added and lysates were immediately transferred to ice to prevent digestion of genomic DNA. Samples were mixed by inverting and the genomic DNA containing precipitate was removed by centrifugation (20,000 x g, 20 min, 4°C). Clarified lysates were extracted twice with a 1:1 mixture of Tris-saturated-phenol (Life Technologies) and $CHCl_3$ (Fisher Scientific), and once with $CHCl_3$ in Heavy Phaselock Gel tubes (5 Prime). Isopropanol (950 µL) was added and the plasmid DNA was pelleted by centrifugation (16,000 x g, 20 min, 4°C), washed twice in 80% ethanol, and resuspended in 200 µL 1x NEB Cutsmart Buffer. Samples were treated with 50 µg/mL RNase A (Life Technologies) at 37°C for 30 min, linearized with PvuII-HF (NEB) at 37°C for 60 min (to aid denaturation during PCR), and treated with 200 µg/mL Protease K at 50°C for 30 min. Finally, each digest was purified using a Zymo gDNA Clean and Concentrator column.

Genomic DNA from MMB-1 strains was extracted using a modified SDS/Protease K method: cells from 300 to 500 µL confluent culture were resuspended in 200 µL lysis buffer (10 mM Tris, 10 mM EDTA, 400 µg/mL protease K, 0.5% SDS) and incubated at 42°C for 1 hr. The digest was purified

using the gDNA Clean and Concentrator Kit (Zymo Research). DNA and RNA preparations were quantified using a Qubit 2.0 Fluorometer (Life Technologies).

## Spacer acquisition assay

Assays were performed as described previously (*Silas et al., 2016*). In figures S1 and S3, we used a Monte Carlo simulation to evaluate a null hypothesis based on random assortment of spacer acquisitions from genomic DNA, with no dependence on gene expression level. Simulations were performed as described previously (*Silas et al., 2016*); for clarity, we provide relevant text from the original method here. For each system, a series of samples of 500 spacers each were randomly chosen in silico from a list of all genes based on the sizes of the individual genes using the stochastic universal sampling algorithm. Sets of 1000 such trials were used to generate a range of null relationships between gene expression and spacer acquisition. The Monte Carlo bounds (black dotted lines on the respective figures) depict the envelope of such simulated random assortments. Traces above this envelope indicate preferential spacer acquisition from highly expressed genes, whereas traces below the envelope indicate spacer acquisition from poorly expressed genes more often than expected by random chance. *M. mediterranea* MMB-1 expression data (at NCBI SRA: SRR2914032, SRR2914033) were previously generated by RNAseq (*Silas et al., 2016*).

## In vivo pre-crRNA processing assay

We used a protocol that preserves strand information, controls for PCR amplification bias, and faithfully reports the 3' ends of source RNA molecules. Total intact RNA (5–10 μg) was run under denaturing conditions on a 6% Novex TBE-Urea polyacrylamide gel (Life Technologies) at 180V for 35 min. Gel fragments corresponding to 30–80 nt size range were excised, and the small RNA fraction was eluted as follows. Gel fragments were shredded and soaked in RNA elution buffer (300 mM NaCl, 1 mM EDTA) overnight at 4°C, and fragments were removed from the eluate by filtration through 0.45 μm Corning Costar Spin-X sterile cellulose acetate filters (Sigma). RNA was precipitated by the addition of 1 ml ethanol, then pelleted by centrifugation (20,000 x g, 20 min, 4°C), and resuspended in RNase-free water. RNA sequencing libraries were prepared as described previously (*Silas et al., 2016*). Sequencing reads containing the first 5 bases of the CRISPR repeat were trimmed, and trimmed reads comprising substrings of the CRISPR repeat sequence were identified. Trimmed reads shorter than 12 bases were also evaluated for a match between the 5 bases preceding the CRISPR repeat and the last 5 bases of genomic CRISPR spacer sequences. Length histograms of trimmed reads in libraries constructed from two independent transconjugant strains are shown for each experiment.

## Plasmid interference assay

The pKT230 vector backbone contains a Kanamycin resistance gene, which allowed us to assay for growth on selective medium as a proxy for CRISPR-Cas interference during the conjugative transfer of the plasmids into MMB-1. A protospacer configuration that allows the plasmid to be targeted by a CRISPR-Cas system leads to plasmid elimination and cell death on selective medium (+Kan). The number of colony forming units per mL of conjugation mixture was determined by dilution and plating on selective medium (2216 marine agar with 50 μg/mL Kanamycin and 50 μg/mL Rifampicin) at 25°C. To account for day-to-day variability in growth between MMB-1 cultures, all protospacer configurations for each host strain were tested on the same day. Plasmid interference in *Serratia* was measured using conjugation efficiency assays, as previously described (*Patterson et al., 2016*). The conjugation efficiency is reported as transconjugants/recipients.

## Relative protospacer transcript level assay

This method is a slight modification of a previously published protocol for CRISPR spacer sequencing (*Silas et al., 2016*); we have provided the protocol in entirety for completeness, retaining relevant text from the original protocol. To standardize the nucleic acid extraction, log-phase cultures of MMB-1 strains carrying pKT230 plasmids with each protospacer configuration (as in *Figure 1—figure supplement 1*) were pooled. RNA was extracted from the pooled samples using TRIZOL reagent and treated with TURBO DNase (Life Technologies) according to manufacturer's instructions. The DNase was subsequently removed by extracting the digests once with a 1:1 mixture of Tris-

saturated-phenol (pH 8.0) and CHCl$_3$, and once with CHCl$_3$ in Heavy Phaselock Gel tubes (5 Prime). Purified RNA (2 μg) was reverse-transcribed into cDNA using Superscript III (Life Technologies), according to the high GC content protocol, with the gene-specific primer AF-SS-425 (TTCAGTTTTC TGATGAAGCGCGAAT). AF-SS-425 binds downstream of the protospacer sequence in a region of the vector backbone common to all plasmids in the mixture. cDNA was treated with RNase H and libraries were prepared for sequencing by a two round PCR method adapted from the spacer acquisition assay. The protospacer locus of the plasmid was amplified from the cDNA sample with primers binding to the same sites in all plasmid configurations, and assayed for the presence or absence of the various protospacer configurations by high throughput sequencing. Round 1 PCR was performed with primers AF-SS-426 (CGACGCTCTTCCGATCTNNNNN AGATGCTGAAGATCAGTTGG) and AF-SS-427 (ACTGACGCTAGTGCATCA AGAAATATCCCGAATGTGCA) as follows: (98℃, 1 min), 2x (98℃, 10 s; 52℃, 20 s; 72℃, 30 s), 16-19x (98℃, 15 s; 65℃, 15 s; 72℃, 30 s), (72℃, 9 min). This simultaneously generated amplicons of near-identical length from all protospacer constructs in a single reaction. Sequencing adaptors were then attached in a second round of PCR with 0.01 volumes of the previous reaction as template, using AF-SS-44:55 (CAAGCAGAAGACGGCATACGAGAT NNNNNNNN GTGACTGGAGTTCAGACGTGTGCTCTTCCGATCACTGACGCTAGTGCATCA) and AF-KLA-67:74 (AATGATACGGCGACCACCGAGATCTACAC NNNNNNNN ACACTCTTTCCC TACACGACGCTCTTCCGATCT) where the (N)$_8$ barcodes correspond to Illumina TruSeq HT indexes D701-D712 (reverse complemented), and D501-D508 respectively. Cycling conditions for this step were: (98℃, 1 min), 2x (98℃, 10 s; 54℃, 20 s; 72℃, 30 s), 3x (98℃, 15 s; 70℃, 15 s; 72℃, 30 s), (72℃, 9 min). Libraries were quantified by Qubit, and sequenced with Illumina MiSeq v3 kits (150 cycles, Read 1; 8 cycles, Index 1; 8 cycles, Index 2).

To verify that all the plasmids in the pooled culture samples used for RNA extraction (as above) were present at comparable levels, a portion of the each sample was reserved for DNA extraction and sequencing. Three mixtures were tested in separate experiments: a mixture of 8 strains in the Cmr2 GGAA mutant background, and two mixtures of 4 strains each in the WT and Cmr4 D26A mutant backgrounds. Relative protospacer RNA incidence is shown as the log$_{10}$ of the number of RNA-derived sequencing reads normalized to the log$_{10}$ of the number of identical DNA reads. This metric is comparable between the various plasmid configurations in a given experiment, but is not a measure of the transcript-to-template ratio in any individual cell because the RNA and DNA libraries were prepared separately and sequenced to similar depths. Controls lacking reverse transcriptase were included, and residual gDNA contamination in DNase treated RNA was determined by PCR titration to be negligible.

## Phage enrichment and isolation

Seawater and sediment samples in contact with *Posidonia oceanica* seagrass were collected from the Mediterranean coast of the Region of Murcia in South Eastern Spain. Samples were kept at 4℃ and processed the day after collection by filtration through cotton cloth, pre-clearing by centrifugation (5000 x g for 10 min at 4℃), and another round of filtration through 0.45 μm mixed cellulose ester filters (Millipore).

Phage amplification was performed with methods adapted from (*Suttle, 1993*). The *M. mediterranea* ΔIII-B strain was used as the host for enrichment from seawater samples to avoid type III-B interference against putative phage infection, and the *M. mediterranea* T103 strain (*Lucas-Elío et al., 2002*) – a histidine kinase mutant that exhibits increased sensitivity to CPG1g infection in addition to other physiological changes – was used as the host for phage enrichment from sediment samples. The host strains were grown in MMC2G broth (NaCl 2%, MgSO$_4$·7H$_2$O 0.7%, MgCl$_2$·6H$_2$O 0.53%, KCl 0.07%, CaCl$_2$·2H$_2$O 0.125%, Peptone 0.5%, Yeast extract 0.1%, Iron (III) citrate hydrate 0.001%, Sodium citrate 0.009%, K$_2$HPO$_4$ 0.4 mM, Glucose 0.2%, pH 7.4). 20 ml of exponential phase cultures were added to 180 ml of seawater samples supplemented with peptone and yeast extract to final concentrations of 0.5% and 0.1% respectively (as in MMC2G). The enrichment cultures were shaken overnight (25℃, 130 rpm). Enrichment cultures were repeatedly sub-cultured in borosilicate tubes until the potential presence of phage was indicated by a decrease in the optical density relative to simultaneous cultures of the ΔIII-B strain without seawater. Finally, the cultures were sterile-filtered and plaque assays were performed with the respective host strains (ΔIII-B or T103) using the double layer agar technique with MMC2 medium (MMC2G with the glucose omitted). The top layer was prepared with 0.6% agar (Pronadisa) and the bottom with 0.8% agar.

## Phage genome sequencing and annotation

The genomes of two phage (CPG1g and CPP1m) that produced plaques of different sizes, and also of the phage isolated from sand samples (CB5A) were sequenced. Phage lysates (500 µL) were digested (10 mM Tris, 10 mM EDTA, 200 µg/mL protease K, 0.5% SDS) at 55°C for 30 min. The digests were extracted once with a 1:1 mixture of Tris-saturated-phenol and $CHCl_3$, and once with $CHCl_3$ in heavy Phaselock gel tubes (5 Prime). DNA sequencing libraries were prepared using the Illumina Nextera kit (Illumina), and sequenced with Illumina MiSeq v3 kits (150 cycles, Read 1; 8 cycles, Index 1; 8 cycles, Index 2). The phage genomes were assembled by SPAdes (3.7.1). Phage genomes were annotated using RAST (http://rast.nmpdr.org/) and PHAST (http://phast.wishartlab.com/). tRNA genes were detected using tRNA-scan SE2 (http://trna.ucsc.edu/cgi-bin/tRNAscan-SE2.cgi). Rho-independent terminators were analyzed using ARNold (http://rna.igmors.u-psud.fr/toolbox/arnold/).

The genomes of phage CPG1g (deposited in GenBank: KY626177) and CPP1m (deposited in GenBank: KY626176) differed at 7 sites: an insertion in position 1809 and 6 single nucleotide variants (SNVs). The insertion generated a frame shift in a 262 aa hypothetical protein (ARB11222) which led to a loss of 27 C-terminal residues (7 residues modified and 20 lost due to premature termination) relative to CPG1g (ARB11272). Only one of 6 SNVs resulted in a change at the protein level: a P448Q variant in the putative DNA primase/helicase (ARB11244 and ARB11284). The genome of phage CB5A was also deposited in GenBank: MF481197.

The 25 remaining phage isolates were tested by PCR to determine if they were likely to be variants of CPG1g. Primers targeting two conserved *Podoviridae* genes were utilized: DpG1gDIR (AACACTTTTAGGATGCGACATAAGT) and DpG1gREV (CCTGTCATCTGCAACAATACATTAAG) amplified a 478 bp portion of the phage DNA polymerase gene, and primers IcG1gDIR (TCACCTCGTGCGATGTTCTC) and IcG1gREV (CATCTCCTCACCTCCATGTTGG) amplified a 728 bp portion of the gene encoding phage protein inside capsid D. All 25 phage isolates produced expected amplicons in both reactions.

## Viral DNA polymerase phylogenetic analysis

Protein sequences similar to the DNA polymerase encoded in the *Marinomonas* phage CPG1g genome were selected by BLAST. Sequences were aligned using the CLUSTAL W algorithm. The MEGA 6 program was used to perform phylogenetic analyses. Trees were constructed using the Neighbor-Joining (NJ) and Maximum Likelihood methods. For the NJ tree, distances between sequences were computed using the p-distance method and are in the units of the number of amino acid differences per site. For the maximum Likelihood (ML) tree, the Le and Gascue substitution model was selected. A discrete Gamma distribution was used to model evolutionary rate differences among sites. Pairwise distances were estimated using a Jones-Taylor-Thorton (JTT) model. All positions with less than 95% coverage were eliminated (i.e. fewer than 5% alignment gaps, missing data, and ambiguous bases were allowed at any position). The reliability of each node in the trees was estimated using bootstrap analysis with 500 replicates.

## *M. mediterranea* IVIA-Po-186 and CPR1 genome sequencing

Isolation of *M. mediterranea* IVIA-Po-186 was described previously (*Espinosa et al., 2010*). *M. mediterranea* CPR1 was isolated from *Posidonia oceanica* plants in Cabo de Palos in July 2005. The roots were washed in sterile saline solution (*Solano et al., 1997*), comminuted, and plated on Marine Agar 2216. Putative *M. mediterranea* strains were identified by their dark pigmentation in this medium. Genomic DNA extracted from IVIA-Po-186 and CPR1 was prepared for high-throughput sequencing using the Nextera DNA Library Prep Kit (Illumina). Contigs were assembled using spades 3.7.1 in 'careful' mode (including read error correction in addition to assembly).

## Evolution experiments to assess inhibition of phage escape by the type III-B system

Log-phase cultures of WT and ΔIII-B strains were inoculated with phage CB5A at MOI of $10^{-2}$, $10^{-4}$, and $10^{-6}$, and grown without shaking at 25°C overnight. All WT infected cultures showed bacterial growth, but only a slight turbidity was observed for the $10^{-6}$ MOI infection of the ΔIII-B strain (whereas $10^{-2}$ and $10^{-4}$ MOI infections were lysed). The $10^{-6}$ MOI infected cultures of both strains

were clarified by centrifugation at 6000 x g, and a 10 µL inoculum of the phage-containing supernatant was used to infect fresh host cells as before. After overnight incubation, the WT infected cultures again showed growth, but the ΔIII-B cultures did not, indicating phage-mediated lysis of host cells. The supernatant was clarified by centrifugation, and phage titers were determined on the WT, ΔIII-B, and ΔI-FΔIII-B indicator strains using the Most Probable Number (MPN) method (*Kott, 1966*). The experiment with the WT host strain was repeated 3 times, whereas 6 replicates of the evolution experiment were performed with the ΔIII-B host. Plaque sizes were also determined on lawns of the ΔIII-B strain, and genomes of 2 phage isolates from each of the 6 independent trials producing large plaques were sequenced as described above.

## Acknowledgements

We thank KL Allison, JA García Chartón, G Mohr, AM Lambowitz, IM García Guillén, and lab colleagues for help and advice. SS was supported by a Stanford Graduate Fellowship and an HHMI International Student Research Fellowship. PCF was supported by a Rutherford Discovery Fellowship from the Royal Society of New Zealand and the Bio-protection Research Centre (Tertiary Education Commission).

## Additional information

### Funding

| Funder | Grant reference number | Author |
| --- | --- | --- |
| National Institutes of Health | R01-GM37706 | Andrew Z Fire |

The funders had no role in study design, data collection and interpretation, or the decision to submit the work for publication.

### Author contributions

Sukrit Silas, Conceptualization, Data curation, Software, Formal analysis, Validation, Investigation, Visualization, Methodology, Writing—original draft, Project administration, Writing—review and editing; Patricia Lucas-Elio, Resources, Data curation, Formal analysis, Validation, Investigation, Visualization, Methodology, Writing—review and editing; Simon A Jackson, Data curation, Formal analysis, Validation, Investigation, Visualization, Methodology, Writing—original draft, Writing—review and editing; Alejandra Aroca-Crevillén, Investigation, Methodology; Loren L Hansen, Data curation, Software, Formal analysis, Investigation; Peter C Fineran, Resources, Data curation, Formal analysis, Supervision, Validation, Investigation, Visualization, Methodology, Writing—review and editing; Andrew Z Fire, Conceptualization, Resources, Data curation, Software, Formal analysis, Supervision, Funding acquisition, Validation, Investigation, Visualization, Methodology, Writing—original draft, Writing—review and editing; Antonio Sánchez-Amat, Resources, Data curation, Formal analysis, Supervision, Validation, Investigation, Visualization, Methodology, Project administration, Writing—review and editing

### Author ORCIDs

Sukrit Silas, http://orcid.org/0000-0003-3251-8579
Patricia Lucas-Elio, http://orcid.org/0000-0001-7182-1189
Simon A Jackson, http://orcid.org/0000-0002-4512-3093
Peter C Fineran, http://orcid.org/0000-0002-4639-6704
Andrew Z Fire, http://orcid.org/0000-0001-6217-8312
Antonio Sánchez-Amat, http://orcid.org/0000-0001-8597-9235

### Decision letter and Author response

Decision letter https://doi.org/10.7554/eLife.27601.030
Author response https://doi.org/10.7554/eLife.27601.031

## Additional files

### Supplementary files

• Supplementary file 1. Summary of all high-throughput sequencing data generated for this study.
DOI: https://doi.org/10.7554/eLife.27601.021

• Supplementary file 2. Sequences of the plasmids used in this study. Names of plasmids used for interference assays specify whether the protospacers match spacers from the type III-B or I-F CRISPR arrays. RC denotes that the reverse complement of a particular configuration was used. DR signifies that the protospacer was flanked by CRISPR repeats of the same array as the spacers.
DOI: https://doi.org/10.7554/eLife.27601.022

• Transparent reporting form
DOI: https://doi.org/10.7554/eLife.27601.023

### Major datasets

The following dataset was generated:

| Author(s) | Year | Dataset title | Dataset URL | Database, license, and accessibility information |
|---|---|---|---|---|
| Silas S, Lucas-Elio P, Jackson SA, Fineran PC, Fire AZ, Sanchez-Amat A | 2017 | CRISPR targeting and spacer acquisition in M. mediterranea mutants, and associated environmental investigations | https://trace.ncbi.nlm.nih.gov/Traces/sra/?study=SRP103952 | Publicly accessible at NCBI Sequence Read Archive (accession no. SRP103952) |

The following previously published datasets were used:

| Author(s) | Year | Dataset title | Dataset URL | Database, license, and accessibility information |
|---|---|---|---|---|
| Silas S, Mohr G, Sidote DJ, Markham LM, Sanchez-Amat A, Bhaya D, Lambowitz AM, Fire AZ | 2016 | total RNA (> 200 nt) sequencing from MMB-1 strains over-expressing RT-Cas1, Cas2, and Marme_0670 - replicate 1 | https://www.ncbi.nlm.nih.gov/sra/?term=SRR2914032 | Publicly accessible at NCBI Sequence Read Archive (accession no. SRR2914032) |
| Silas S, Mohr G, Sidote DJ, Markham LM, Sanchez-Amat A, Bhaya D, Lambowitz AM, Fire AZ | 2016 | total RNA (> 200 nt) sequencing from MMB-1 strains over-expressing RT-Cas1, Cas2, and Marme_0670 - replicate 2 | https://www.ncbi.nlm.nih.gov/sra/?term=SRR2914033 | Publicly accessible at NCBI Sequence Read Archive (accession no. SRR2914033) |

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
