## [Decision Letter]

Thank you for submitting your article "Mobile type III CRISPR-Cas systems can provide redundancy to counteract viral escape from type I systems" for consideration by *eLife*. Your article has been reviewed by three peer reviewers, one of whom served as Guest Reviewing Editor, and the evaluation has been overseen by Gisela Storz as the Senior Editor. The following individual involved in review of your submission has agreed to reveal his identity: Edze Westra (Reviewer #1).

The reviewers have discussed the reviews with one another and the Reviewing Editor has drafted this decision to help you prepare a revised submission.

Summary:

Work by these authors previously showed that the Type III-B system in M. mediterranea MMB-1 is capable of acquiring new spacers from RNA by using an RT-Cas1 fusion protein. Here they extend this work to address questions related to the orientation of newly integrated spacers, the mechanism(s) of target degradation by the I-F and III-B systems, the promiscuity of sharing spacers between these systems, and speculate on the selective pressures that would explain the co-occurrence of these two systems. The referees agree that the work is generally well-performed but the novelty is substantially overstated.

Essential revisions:

1) Claims that the work presents unique and new insights into CRISPR biology are largely overstated and these claims should be revised to more accurately reflect the contribution of previous work.

2) The rather poor immunity provided by the type III-B system when using the type I-F spacer, indicates that type III-B-mediated interference using a type I-F-derived spacer is impaired. It is critical that type III-B-mediated phage defense is measured with spacers derived from its own type III-B array. In addition, RNA-seq from pull-down type I-F and III-B complexes should be used to corroborate that type I-F crRNAs are also present in the type III-B complex but not vice versa.

3) The authors state that the type III-B system provides an additional layer of protection. However, all experiments are performed using fixed genotypes. To demonstrate that type III limits phage escape, we recommend evolutionary experiments. Phage challenge of cells with a wild type immune system (i.e. I-F and III-B), as well as the isogenic mutant lacking III-B, would allow the authors to test the hypothesis by looking for phage escape mutants after 1, 2 and 3 days.

4) The authors describe a scenario that might explain the selective force that caused the III-B and I-F systems to end up in the same host. However, the sparse genomic data used is not convincing and the proposed scenario seems contrived. This section of the paper should be moved to the Discussion.

5) Repeat spacer analysis in Figure 2 with spacers acquired from plasmids.

Title:

The title is inaccurate because there is little proof of the mobility of the studied type III-B system. The title is also confusing because it gives the impression that the investigators are reporting experiments about the transfer of the type III system, for example via conjugation or another form of gene transfer route.

---

## [Author Response]

*Essential revisions:*

*1) Claims that the work presents unique and new insights into CRISPR biology are largely overstated and these claims should be revised to more accurately reflect the contribution of previous work.*

We have carefully revised the text to ensure that the novel aspects of our study are clear in the context of previous work in the field. Specifically, identifying a deficiency in type III spacer acquisition (the apparent lack of a mechanism to correctly orient new spacers in the CRISPR loci), the plasticity of crRNA utilization by type III interference machinery (as it is able to utilize crRNAs with diverse CRISPR repeats), the biological function of crRNA-stealing (or sharing) among cohabiting CRISPR-Cas systems (including type III-style self- non-self- discrimination with type I-F crRNAs), and the physiological consequences of this in the native host ecosystem (providing defense against phage that escape type I-F immunity and suppressing proliferation of such mutants).

In terms of the specific history of utilization of the same crRNAs by different CRISPR-Cas systems, we have expanded our introductory discussion of several studies in which co-precipitation analyses suggested some cross-association between RNAs and various interference complexes (Staals et al., 2013; Staals et al., 2014; Elmore et al., 2015; Majumdar et al., 2015), and the studies by Deng et al., 2013 and Elmore et al., 2016 that provided the first evidence of functional “crRNA sharing” from “communal” arrays using engineered plasmids. These indications spark considerable interest in the possibility that such immunological redundancy might be effective in a natural ecosystem. With some new data, we provide explicit evidence (below) that promiscuous crRNA use by the type III-B system in *M. mediterranea* MMB-1 provides PAM-independent backup to type I-F CRISPR-Cas immunity against naturally occurring phages that have PAM mutations allowing them to escape type I-F interference. We hope the revised manuscript provides a clearer description of the unique new insights of our work in the context of the existing literature.

We have also clarified another significant aspect of the cooperation between CRISPR systems, related to the architecture of the CRISPR-Cas systems. Type III systems (such as the type III-B loci in *S. islandicus* REY15A used by Deng et al., and the type III-B loci in *P. furiosus* used by Elmore et al.) often lack CRISPR arrays, as well as components for acquisition of spacers (i.e. *cas1* and *cas2*) and for crRNA processing and maturation (i.e. *cas6*). The earlier data were consistent with models (certainly still plausible) that different systems co-habiting the same host might potentially share ‘communal’ CRISPR arrays, with the systems having co-evolved to use identical CRISPR repeat sequences for crRNA selection and loading onto the interference machinery. In contrast, the type III-B system in *M. mediterranea* MMB-1 has evidently been recently acquired (relative to the type I-F system that is conserved in very closely related *M. mediterranea* strains, whereas the type III-B systems differ both in sequence and genomic context) and contains its own components for each step of the CRISPR immunity pathway, including CRISPR arrays that we demonstrate are specific to the type III-B system (yielding precursor crRNAs that can only be processed by type III-B factors). The observation that the *M. mediterranea* MMB-1 type III-B system can also use type I-F crRNAs, shows an opportunistic and potentially adaptive flexibility to both own and borrow guides, and establishes an entirely new model of plasticity in crRNA selection by the type III interference machinery.

The key points here have been substantially strengthened in the revised manuscript through the isolation and sequencing of an additional phage, and through characterization of the corresponding infectivity patterns in laboratory evolution experiments. We hope that we have now better articulated how these findings fit within, and extend, the context of previous work – in particular, the previously enigmatic co-occurrence of type III and type I systems.

Please also refer to our response to point 4.

*2) The rather poor immunity provided by the type III-B system when using the type I-F spacer, indicates that type III-B-mediated interference using a type I-F-derived spacer is impaired. It is critical that type III-B-mediated phage defense is measured with spacers derived from its own type III-B array. In addition, RNA-seq from pull-down type I-F and III-B complexes should be used to corroborate that type I-F crRNAs are also present in the type III-B complex but not vice versa.*

We now explain this in more detail in the paper. Multiple lines of evidence support a substantially effective immune response by the type III-B system. This should now be clearer, both from more explicit explanation and some additional data in the paper.

Immunity can be measured on a variety of scales, and the immunity provided by the type III-B system using the type I-F spacers is substantial in each of the individual assays. Our plasmid inference assays provide a direct comparison for the type III-B system using its own spacers versus type I-F spacers. In both cases we observed at least a two order-of-magnitude decrease in plasmid conjugation efficiency. (E.g. compare results of the plasmid interference assay with the ∆I-Fcas123 strain (only type III interference possible in this strain) in Figure 1 (type III interference using spacers from the type III-B array) and Figure 3 (type III interference using spacers from the type I-F array). To make the corollary argument: that anti-plasmid type I and type III interference efficiencies are similar, compare type III interference using type I spacers in the ∆I-Fcas123 strain (only type III interference possible, Figure 3) with PAM-enabled type I interference using type I spacers in the ∆III-B strain (only type I interference possible, Figure 5)). Because different spacer sequences provide differing levels of immunity, even within the same system (e.g. Xue, NAR 2015), we had performed the experiments in Figure 1, Figure 3, and 5C-D with two examples of each spacer type.

In regards to phage defense, although the ~10 fold increase in CPG1g phage EOP observed in the absence of the type III-B system might appear modest, this is still greater than the immunity provided by the type I-F system (owing to the PAM and protospacer mutations in the phage) – representing a fitness advantage for strains possessing the type III-B system (see also our response to the reviewers’ point 4). Moreover, plaque formation is a complex process dependent on several factors (e.g. the host growth rate, phage burst time and size) and the absolute difference in plaque numbers (EOP) provides only one metric of immunity. As such, we now emphasize both the EOP and plaque size (e.g. larger plaques are formed in the absence of type III-B immunity, Figure 3).

An additional set of arguments comes from our isolation of a new phage, CB5A, which is targeted by the *M. mediterranea* MMB-1 type I-F system with a canonical PAM (GG) (see also our response to reviewers’ point 3). This has enabled us to present a broader examination of the role type I-F spacers play in phage immunity, and to directly address the reviewers’ concern that type III immunity relative to type I anti-phage defense is somehow impaired. When we examined efficiency of plaquing for CB5A, we observed that knockout of the type III-B system increases the EOP >1000 fold, whereas the knockout of type I-F interference (the *cas1, cas2-3* deletion strain) only increases the EOP ~50 fold (note that the type I-F system can only use one spacer to target CB5A, compared with multiple type I-F derived spacers used by the type III-B system). The differences between CB5A and CPG1g infectivity (Figure 3 and Figure 7) in the presence or absence of various CRISPR-Cas components highlight that the type III-B system provides robust anti-phage defense using type I-F crRNAs, and likely reflect the evolutionary divergence of these phage, driven by host CRISPR-Cas immunity (see also the expanded Results section “The type III-B CRISPR-Cas system counteracts viral escape from the type I-F system”).

In response to the second point of the reviewer’s comment (that RNAseq from pull-downs of the type I-F and type III-B complexes should be performed) we feel that although it might be nice to include complementary data such as this, it would not provide any additional biological insights beyond what we have extensively demonstrated with functional assays in vivo. Given the strong functional evidence (Figure 1, Figure 3, Figure 5; of particular note, assays indicating that type III interference using type I-F spacers requires multiple type III-B components including Cmr2, Cmr4, and the GGDD motif in Cmr2) and challenges in interpreting co-precipitation data (e.g. non-specific RNA-protein binding could occur even in the absence of a functional interaction; conversely,the lack of an association in a co-immunoprecipitation assay would not provide a reliable assessment of the lack of an interaction in vivo), we consider these experiments to be beyond the scope of this study.

*3) The authors state that the type III-B system provides an additional layer of protection. However, all experiments are performed using fixed genotypes. To demonstrate that type III limits phage escape, we recommend evolutionary experiments. Phage challenge of cells with a wild type immune system (i.e. I-F and III-B), as well as the isogenic mutant lacking III-B, would allow the authors to test the hypothesis by looking for phage escape mutants after 1, 2 and 3 days.*

We share the reviewer’s enthusiasm for these laboratory-evolution type experiments. We were initially precluded from performing these, due to the lack of a phage-host combination where the host possessed type I-F–specific immunity against the phage (due to the PAM mutations in CPG1g). We investigated ‘engineering’ a relationship in our phage-host setup, but this would be impractical. Fortunately, we were able to address this point by isolating another new phage from the marine environment. We performed additional phage isolations from sediment samples obtained from Mediterranean sea-grass meadows in Calabardina (Aguilas, Spain) using a previously isolated *M. mediterranea* strain with a histidine kinase mutation that renders the host more sensitive to CPG1g infection (in addition to altering the regulation of other physiological processes such as melanin synthesis). The new phage, designated CB5A, is closely related to CPG1g and may more closely represent an ancestor of phage CPG1g. This phage contains an intact PAM next to a protospacer that perfectly matches the second native spacer in the type I-F CRISPR04 array in the MMB-1 genome. The intact PAM implies that this protospacer would be recognized by both the type I-F and III-B systems. Consistent with this expectation, it produces small plaques on lawns of the ∆I-F Cas123 strain (i.e. lacking type I-F interference), or the ∆III-B strain (lacking type III-B interference; note that CPG1g – an “escape” phage itself – produces large plaques on this strain (Figure 3)). Phage CB5A was passaged on WT and ∆III-B strains with the addition of fresh host cells daily at each sub-culturing step, and mutant phage escaping the type I-F system arose after just one re-inoculation on the ∆III-B host, but not with the WT – providing evidence that the ability of the III-B system to use I-F crRNAs reduces the proliferation of “escape” phages.

Eleven CB5A escape mutants from 6 independent evolution experiments were isolated and characterized by whole genome sequencing: all contained deletions or mutations of the PAM-containing protospacer. These phage isolates were also shown to match the CPG1g “escape” plaquing phenotype (i.e. large plaques on the ∆III-B host) instead of the CB5A phenotype (small plaques on the ∆III-B host). These results are described in the new Figure 7 and its supplements, and are congruent with our model that the type III-B system helps to counter phages that escape type I-F defenses, thereby providing a fitness advantage to the host. We thank the reviewers for suggesting we pursue this experiment as we consider that this has added significant evolutionary context for our findings.

*4) The authors describe a scenario that might explain the selective force that caused the III-B and I-F systems to end up in the same host. However, the sparse genomic data used is not convincing and the proposed scenario seems contrived. This section of the paper should be moved to the Discussion.*

We thank the reviewers for the feedback on this section. It is our view that the genomic sequencing and comparative genomics data in this section are a critical part of the narrative. We have also clarified this in the reworked discussion of this section.

To better convey the close genetic relationship between the *M. mediterranea* strains that we sequenced as part of this study, we have performed average nucleotide identity (ANI) analyses to definitively establish relationships between these isolates of *M. mediterranea.* From this analysis, it is clear that theseare very closely related strains of the same species that exhibit remarkable differences between the type III CRISPR-Cas loci that they possess and/or lack. One of the key differences between our study and previous work, is that past studies (e.g. Deng et al., 2013, Elmore et al., 2016) suggest a model where type III systems stably co-reside with type I loci to provide redundancy or prevent evasion by anti-CRISPRs or phage escape mutants. We demonstrate that type III systems can indeed provide immunological redundancy in nature, with the genomic data presented in this section of our paper leading strongly to a model in which type III-B systems in *M. mediterranea* were more recently acquired than the type I-F system (which is conserved in all sequenced *M. mediterranea* strains). Thus, a more dynamic situation occurs in natural populations of *M. mediterranea*, whereby uptake and/or loss of diverse type III modules could be selected for by pressures arising in the course of the phage-host arms race. We propose that the apparent horizontal mobility of type III CRISPR interference modules even in closely related bacterial genomes (shown here for *Marinomonas* species, and also previously observed for strains of *S. islandicus* by Held et al., 2013) is advantageous due to their ability to use crRNAs from type I CRISPR loci, rather than a sign of their dependence on type I systems. In addition, as detailed in our earlier responses, our new data provide experimental evidence that the MMB-1 type III-B system suppresses proliferation of phage mutants that might otherwise have escaped type I-F defenses – supporting a selective advantage for III-B systems through co-opting type I-F crRNAs.

*5) Repeat spacer analysis in Figure 2 with spacers acquired from plasmids.*

We have performed these analyses as recommended. Despite the acquired spacer counts from plasmid being low, the binary relationships between the sense and antisense fractions were maintained in all experiments (with one exception, ∆Cmr4, likely a statistical anomaly from the low plasmid-derived-spacer counts). As nothing unexpected arose from this analysis, we would prefer to omit these results from the figures to avoid potential, unnecessary overwhelming of readers. See Author response image 1.

**Author response image 1. respfig1:** (As in Figure 2) Proportion of newly acquired spacers isolated from CRISPR03 mapping to sense and antisense strands of plasmid-borne genes.

*Title:*

*The title is inaccurate because there is little proof of the mobility of the studied type III-B system. The title is also confusing because it gives the impression that the investigators are reporting experiments about the transfer of the type III system, for example via conjugation or another form of gene transfer route.*

We are grateful to the reviewer for this observation. Certainly, the intent was not to mislead the readership as to the nature of experiments performed. We have modified the title accordingly.